# Rh, Ir, and Ru Partitioning in the Cu-Poor IPGE Massive Ores, Talnakh Intrusion, Skalisty Mine, Russia

**Nadezhda Tolstykh** [1,*], **Valeriya Brovchenko** [2], **Viktor Rad'ko** [3], **Maria Shapovalova** [1], **Vera Abramova** [2] **and Jonathan Garcia** [1,4]

1    Sobolev Institute of Geology and Mineralogy, Siberian Branch of the Russian Academy of Sciences, 630090 Novosibirsk, Russia; shapovalovam@igm.nsc.ru (M.S.); jonathan.andres.garcia@gmail.com (J.G.)

2    Institute of Geology of Ore Deposits Mineralogy, Petrography and Geochemistry, Russian Academy of Sciences, 119017 Moscow, Russia; valeriiabrovchenko@gmail.com (V.B.); winterrain@rambler.ru (V.A.)

3    OOO "Norilskgeologiya", 663300 Norilsk, Russia; radkovik@yandex.ru

4    Department of Geology and Geophysics, Novosibirsk State University, 630090 Novosibirsk, Russia

\*    Correspondence: tolst@igm.nsc.ru

**Abstract:** Pyrrhotite (or Cu-poor) massive ores of the Skalisty mine located in Siberia, Russia, are unique in terms of their geochemical features. These ores are Ni-rich with Ni/Cu ratios in the range 1.3–1.9 and contain up to 12.25 ppm Ir + Rh + Ru in bulk composition, one of the highest IPGE contents for the Norilsk–Talnakh ore camp. The reasons behind such significant IPGE Contents cannot simply be explained by the influence of discrete platinum-group minerals on the final bulk composition of IPGE because only inclusions of Pd minerals such as menshikovite, majakite, and mertieite II in Pd-maucherite were observed. According to LA-ICP-MS data obtained, base metal sulfides such as pyrrhotite, pentlandite, and pyrite contain IPGE as the trace elements. The most significant IPGE concentrator being Py, which occurs only in the least fractionated ores, and contains Os up to 4.8 ppm, Ir about 6.9 ppm, Ru about 38.3 ppm, Rh about 36 ppm, and Pt about 62.6 ppm. High IPGE contents in the sulfide melt may be due to high degrees of partial melting of the mantle, interaction with several low-grade IPGE impulses of magma, and (or) fractionation of the sulfide melt in the magma chamber.

**Keywords:** Talnakh intrusion; Skalisty mine; pyrrhotite Cu-poor ores; Rh; Ir; Ru-rich pyrite; Pd–Ni arsenides

## 1. Introduction

The formation and evolution of the Cu–Ni massive ores of the Norilsk–Talnakh ore district, located on the northwestern edge of the Siberian Platform in the Russian Arctic, have always excited scientific interest as to the development of genetic models and prospecting strategies [1,2]. The latter is because some of the intrusive bodies associated with flood-basalt magmatism represent one of the largest accumulations deposits of magmatic sulfides enriched in platinum group elements (PGE). Talnakh and Kharaelakh intrusions are good illustrations of such anomalous enrichments; additionally, these ore bodies are spatially confined to the main Norilsk–Kharaelakh fault (Figure 1a) and their massive sulfide ores of up to 45 m thick have brought important scientific and industrial attention.

Sulfide deposits of the Norilsk region, their genesis, and mineralogical– geochemical features have already been extensively studied for many years [2–21]. Regardless of this, the Norilsk Cu–Ni–PGE deposits remain as unique geological objects, which have yet not found common consensus within all the spectrum of hypothesis regarding their origin and evolution.

All ore-bearing intrusions are composed of the same type of rocks (gabbroic) in a broad sense and are represented by chonolithe, ribbon-like- and trough-like-body structures. The section of the intrusion (Figure 2) from bottom to top comprises contact and taxitic (or olivine) gabbro–dolerites of the "lower gabbro series"; followed by picritic, olivine,

olivine-bearing and olivine-poor gabbro–dolerites of the "main layered series", in the middle part of the intrusion; then the "upper gabbro series"; and the endocontact zone of the intrusion composed by taxitic and leucocratic gabbros including the low-sulfide horizon [1,9,10,13,16,20].

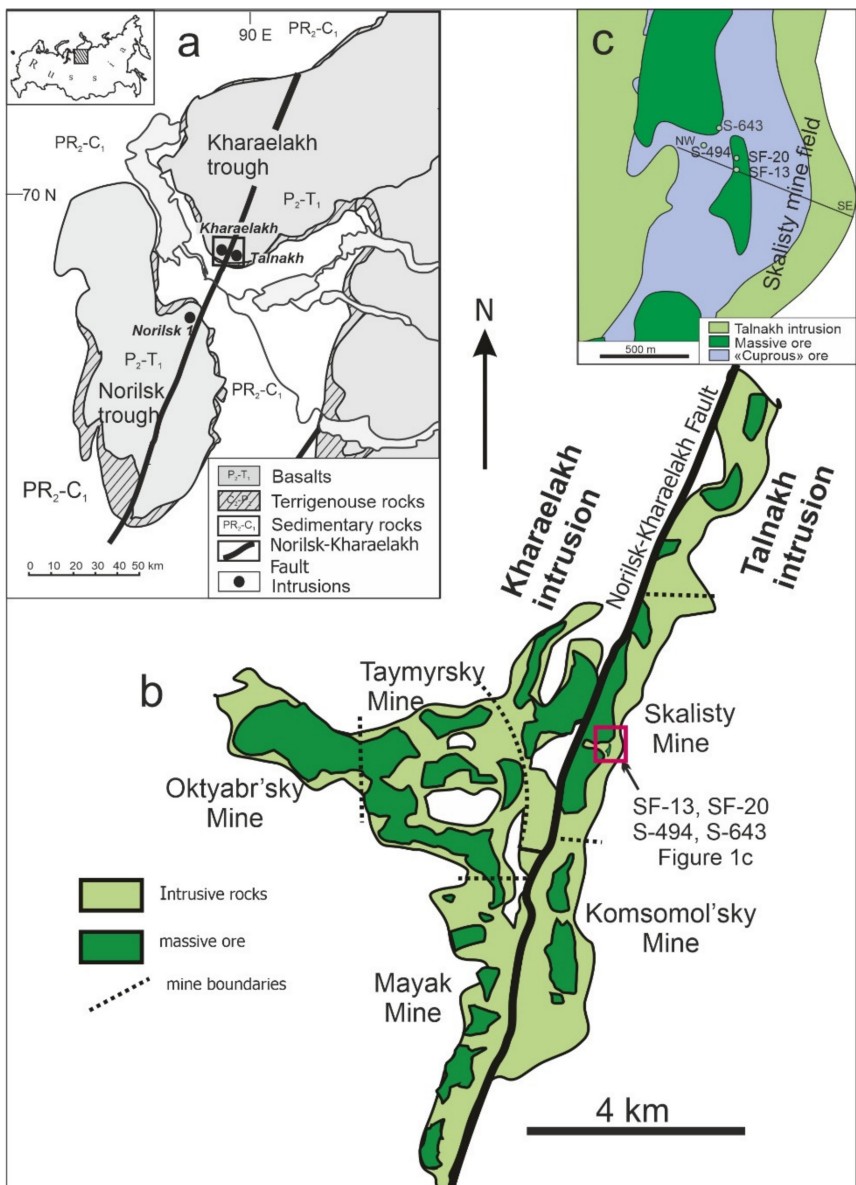

**Figure 1.** Schematic geological map of the Norilsk–Talnakh district (**a**) and surface projection of the Talnakh and Kharaelakh intrusions modified from [33], including the location of the Skalisty mine (**b**) and the sampling location (**c**).

Different types of ores have been described already [1,12] for the Norilsk–Talnakh ore district (Norilsk 1, Oktyabrsky, and Talnakh deposits): (1) disseminated ores in the lower part of intrusions hosted by picritic, taxitic, or olivine gabbro–dolerites (Main Ore Horizon) according to [22], picritic gabbro-dolerite classifies as an olivine-rich orthocumulate with >20% high-Mg olivine and >18 wt.% MgO in whole-rock; and taxitic (or olivine) gabbro–dolerite corresponds to a texturally heterogeneous gabbro or olivine gabbro with extensive grain size variability; (2) massive ores that typically occur in the lower endocontact of intrusion or in the host rocks (exocontact); (3) disseminated and vein-disseminated "cuprous"

ore developed in hanging-wall positions to the intrusions, commonly within hornfels [22]; (4) a low sulfide horizon at the base of the upper gabbro series in taxitic leucocratic gabbro.

Massive ores can be Cu-poor (pyrrhotite) or Cu-rich (chalcopyrite or mooihoekite type), often exhibiting spatial zoning. The Kharaelakh intrusion hosts the orebody of the Oktyabrsky deposit, which shows zoning from pyrrhotite to cubanite and mooihoekite or talnakhite ores. Its detailed mineralogy has been investigated and discussed in several works (e.g., [4–7,23,24]).

The most common association is the one comprising pyrrhotite–chalcopyrite ores ranging from 10 to 15 ppm PGE with a predominance of palladium over platinum [15]. These ores may also exhibit distinct sulfide zoning varying from pyrrhotite-dominated to progressively more Cu-rich zones that are commonly more enriched in PGE. This zoning is attributed to in situ fractionation of sulfide liquid [1]. A noteworthy example from these massive ores, consider to be one of the most enriched in chalcopyrite ores, is the Southern-2 orebody (PGE content of 220 ppm) occurring in the southwestern branch of the Talnakh intrusion [17,25]. The formation of such ores can be interpreted applying the following model: a large volume of immiscible copper-rich sulfide liquid fractionated in an intermediate-depth magma chamber, later this liquid was entrained together with a small volume of silicate magma into the sedimentary rocks at the bottom of intrusions [8,26–28]. The Cu-rich ores are enriched in chalcophile elements Te, As, Bi, Sb, and Sn [29], and on top of this contain minerals of the platinum group elements (PGMs) crystallizing in different systems: Pd–Sn, Pd–Bi–Te, Pd–Sb–As, and Pd–Pb, often associated with Au–Ag alloys [21].

On the other hand, the pyrrhotite massive ores are thought to represent a weakly fractionated portion of the sulfide melt and are found widespread in all ore-bearing intrusions of the Norilsk region. Additionally, they have been identified in the northeastern branch of the Talnakh intrusion (Komsomolsky and Skalisty mines) and consider of less economic interest [15]. However, according to the study of [30] the pyrrhotite massive ores from the Skalisty mine are unique as to their content of elements of the platinum group of the iridium subgroup—Ir, Ru, Os (IPGE) and Rh, which have also been previously confirmed by [31]; thus, reconsideration of its economic significance and further investigation should be contemplated.

Generally, pyrrhotite ores are poor in terms of PGM species. The most common minerals in the pyrrhotite ores of the Norilsk-1 intrusion are sperrylite and tetraferroplatinum; in relation to Pd minerals in pyrrhotite ore, these are mainly represented by two systems: Pd–Sn and Pd(Ni)–As [9]. However, sobolevskite–kotulskite solid solutions; Rh-, Pt-, Ir-sulfoarsenides; and laurite have also been found in these types of ores [32].

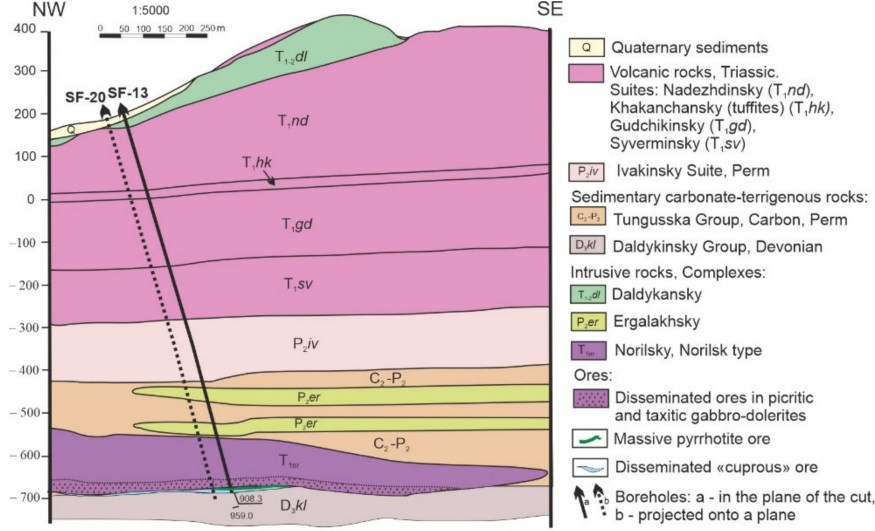

**Figure 2.** Geological cross-section of the Talnakh intrusion (northwestern branch) showing the position of the SF-13 and SF-20 boreholes.

Thus, pyrrhotite ores of the eastern flank in the Skalisty mine, Talnakh intrusion, may be recognized as unusual, regarding their atypical enrichment in Rh, Ru, and Ir [2,30,31]. Similarly, in pyrrhotite ores from the core of the chonolith (FS-13 and SF-20), PGMs are completely absent; therefore, the anomalous Contents of IPGE should have been allocated (completely dissolved) in any of the present sulfides and not as discrete PGM phases. Whereas ores from the soul bands of the orebody contain only Pt–Pd minerals, mainly in the Pd–Ni–As systems [31]. A rigorous analysis of the existing hypothesis on the processes affecting the Content and distribution of chalcophile elements in base metal sulfides (BMS) is provided in [34] and references therein.

The purpose of our research is to (1) identify the sulfides that could have dissolved such significant amounts of Rh and IPGE, and (2) to understand the mechanisms of enrichment of certain sulfides in these elements, in ores formed from sulfide melts with variable degrees of fractionation.

## 2. Materials and Methods

The chemical composition of minerals was investigated at the Analytical Center for Multielement and Isotope Research in the Sobolev Institute of Geology and Mineralogy of the Siberian Branch of the Russian Academy of Sciences (analyst M. Khlestov), both as heavy concentrates of crushed samples mounted into polished sections and as thin sections of rocks using the Microanalyzer Mira 3 (Tescan, Czech Republic) with an energy dispersive X-ray (EDS Oxford X-Max 80) spectrometer with the following operating parameters: accelerating voltage 20 kV, probe current 1.6 nA, spectrum acquisition time on samples 20 s, and beam size of <0.1 μm. The following standards: pyrite (S), $PtAs_2$ (As), HgTe (Te), metallic Fe, Co, Ni, Cu, Pt, Pd, Au, Ag, Sb, Bi, and Pb, were used for this purpose. The detection limit for elements was 0.2–0.3 wt.%. Correction for matrix effects was performed using the XPP algorithm.

The WDS analyses of BMS were carried out using a Camebax-Micro electron probe microanalyzer (EPMA), Cameca Ltd., Gennevilliers, France, (~1 mm beam) at 20 kV voltage, 20–30 nA current, and 10 s measurement for each analytical line. The following standards were used: pure metals for Au and Ag during the analysis of Au–Ag alloys; synthetic compounds $CuFeS_2$ for Cu, $FeS_2$ for Fe and S, and Ni, FeNiCo for Co, PbS for Pb.

The following X-ray lines were selected: $K\alpha$ for S, Fe, Cu, Ni and Co; $M\alpha$ for Pb. Overlapping of elements in the X-ray spectra was corrected with the assistance of a program including experimentally calculated coefficients [35]. The accuracy and reproducibility of analytical methods and the comparison between WDS and EDS data were assessed with special tests [36,37]. Additional elemental analysis was performed via the JXA-8100 (JEOL Ltd., Tokyo, Japan) (wt.%) with detection limits: Cu 0.016, Ni 0.007, Co 0.007, Pb 0.012, S 0.007, Fe 0.012. The amount of each element was calculated using the XPP software. The accuracy and reproducibility of the results were evaluated in the same way as in the previous methods.

EDS elemental distribution maps were obtained using an EPMA JXA-8100 (JEOL Ltd., Tokyo, Japan). The analytical assessment was performed at V = 20 kV, I = 100 nA, and time (*t*) to set each pixel in the map, 0.2 s. The color scale is generated automatically by the microanalyzer software and reflects the relative Contents of a chemical element using 256 colors. The scale may show different units relative to the minimum and maximum Contents of the element, which can be distinguished by the microanalyzer.

The Contents of platinum-group elements ($^{99}$Ru, $^{101}$Ru, $^{103}$Rh, $^{105}$Pd,$^{108}$Pd, $^{185}$Re, $^{189}$Os, $^{191}$Ir, $^{193}$Ir, $^{195}$Pt) and $^{33}$S, $^{55}$Mn, $^{59}$Co, $^{57}$Fe, $^{60}$Ni, $^{61}$Ni, $^{63}$Cu, $^{65}$Cu, $^{75}$As, $^{107}$Ag, $^{109}$Ag, $^{125}$Te, $^{128}$Te, $^{182}$W, $^{197}$Au, $^{205}$Tl, $^{208}$Pb, and $^{209}$Bi, in sulfides, were determined via laser ablation inductively coupled plasma mass spectrometry (LA-ICP-MS) at IGEM-Analitika in the Institute of Geology of Ore Deposits Mineralogy, Petrography and Geochemistry, Russian Academy of Sciences. The LA-ICP-MS system consists of a New Wave UP-213 solid-state Nd:YAG laser coupled with a Thermo XSeries2 quadrupole ICP-MS. For external calibration, the standard reference materials (RSM) were UQAC FeS-1 (University of

Quebec, Chicoutimi, Canada), which are manufactured from natural sulfide powder and doped with trace elements; additionally, MASS-1 [38]—a ZnCuFeS pressed powder pellet provided by the US Geological Survey—was used to verify the results and for some missing elements in the UQAC FeS-1. The results of the RSM measurements of all sessions and the detection limits are given in Appendix A Table A1.

LA-ICP-MS analyses were carried out using a 40–60 μm beam in diameter for spots and 30–40 μm for profiles. A laser frequency 15 Hz, 5–7 J/cm$^2$ energy density and 7 μm/s speed of moving the laser beam on top of the sample surface. The carrier gas was a mixture of helium (0.7 L/min) and argon (0.85 L/min). The acquisition time in spot mode was 60 s for mineral analysis and 30 s for washout. Each linear profile was preceded by 30 s of background. Signal quantification was processed by Iolite software [39] using $^{57}$Fe content in each sulfide as determined by the electron microprobe as an internal standard. Profiles of intergrown minerals were assessed separately, avoiding areas around contacts. Where microinclusions of platinum-group minerals (PGM) were identified by the LA-ICP-MS spectra, the corresponding part of the spectrum was cutout offline. The interference of zinc argide $^{68}$Zn$^{40}$Ar in $^{108}$Pd, and Cu argides $^{63}$Cu$^{40}$Ar and $^{65}$Cu$^{40}$Ar interferences in $^{103}$Rh and $^{105}$Pd, respectively, were monitored by measuring MASS-1 as a blank, which does not contain Ru, Rh, and Pd, but contains Zn and Cu as major components. In pentlandite, $^{99}$Ru was used instead of $^{101}$Ru. Based on the values obtained in the blank, 0.5 ppm $^{103}$Rh was subtracted for every 1 wt.% Cu from each mineral result. In addition, $^{108}$Pd was used instead of $^{105}$Pd to avoid interferences in Cu-rich minerals. The detection limits for LA-ICP-MS analyses were calculated as three sigma (σ) times the background counts for the gas blank.

## 3. Results

### 3.1. Mineralogical Features and Textures of the Massive Ores

Samples of IPGE-rich massive ores from boreholes SF-20 and SF-13 exhibit similar textural and structural features (Figure 3). Textures are as a norm massive (92–99 vol.% of sulfides), and structures are predominantly granular. Pyrrhotite prevails among the ore minerals, reaching up to 75 vol.% for these samples. Chalcopyrite clusters of isometric or veinlet shapes are usually located within a pyrrhotite matrix.

Large pyrite grains occur forming chains or tails of individual crystals along the margin of chalcopyrite segregations (Figure 3a), SF-13 sample. Pyrite is normally restricted to the border of chalcopyrite and never crosses or penetrates it (Figure 3b,c). Pentlandite in these ores appears subordinate, represented by three modifications: thin lamellas or dendritic crystals in pyrrhotite (MSS exsolution) (Figure 3d,f–h); thin rim-like precipitates at the contact between pyrrhotite and chalcopyrite (Figure 3a,b); and ribbon-like accumulations of grains confined to chalcopyrite clusters (Figure 3g,i). Magnetite occurs as isometric and rounded inclusions, and sometimes subhedral, restricted to chalcopyrite–pentlandite segregations (SF-20).

Massive ores from borehole S-643 occurring at the margins of the main massive ore body have textural and structural characteristics that differ from those described in the aforementioned boreholes. In these ore samples, chalcopyrite plays a more significant role in the balance of ore minerals (Figure 4a–e). Pentlandite appears forming dendritic-like "veins" in pyrrhotite and chalcopyrite grains. For these ores, Pn lamellae in Po are less common, i.e., pyrrhotite grains without such exsolution textures are present. A typomorphic feature of this sample is the presence of zoned magnetite grains, the rim of which is represented by ilmenite (Figure 4a) or ilmenite–magnetite intergrowth (Figure 4d,e). Ilmenite also occurs commonly as large tabular crystals (Figure 4b) including abundant titanite and perovskite lamellae. Droplet-like sulfide inclusions in oxide minerals were found in samples from this borehole (Figure 4b,e).

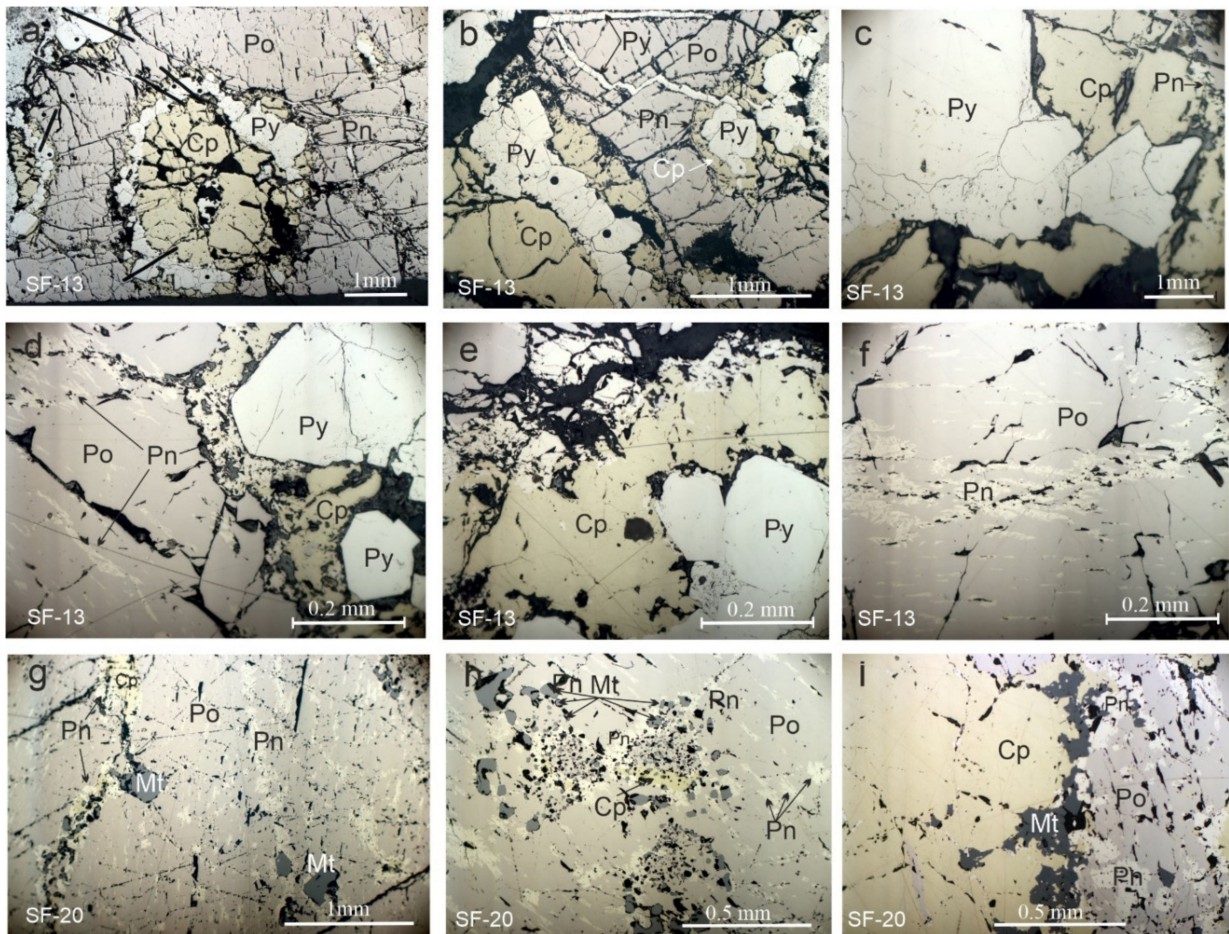

**Figure 3.** Textures and parageneses of sulfides in massive ores from boreholes SF-20 (**a**–**c**), and SF-13 (**d**–**i**), reflected light. (**a**–**c**) Zoned sulfide segregations (**a**), in pyrrhotite (Po): a core composed of chalcopyrite (Cp) along the margin of which occur granular pyrite (Py), and a thin pentlandite (Pn) rim along the chalcopyrite (**b**,**c**) within a pyrrhotite matrix; (**d**–**f**) sulfide parageneses in which Pn is represented by two generations: lamellae of exsolution in Po (**d**,**f**) and rims along the border of chalcopyrite segregation (**d**,**e**); (**g**) lamellae of pentlandite (Pn) and veinlets of magnetite (Mt)–Cp–Pn composition in Po; (**h**) segregation of the Mt–Cp–Pn composition in Po; (**i**) Mt–Po–Cp–Pn intergrowth.

Samples from Borehole S-494, located farther north at the margin of the massive orebody, are also IPGE-enriched as SF-13 and SF-20 samples; pyrrhotite and chalcopyrite occur in equal proportions (Figure 4f). Chalcopyrite sometimes may prevail compared to other sulfides and appears as a massive matrix for other sulfide (Po, Pn, and Mt). On the other hand, chalcopyrite segregations, as a rule, are framed by rim-like pentlandite (Figure 4g–i). Segregations of the Mt–Cp–Pn association in Po are common.

### 3.2. Composition of the Ore Minerals from the Massive Ores

The composition of pyrrhotite varies in a range $(Fe,Ni)_{0.86-0.87}S_{1.00}$ or $Me_7S_8$ (Me = Fe + Ni) for samples SF-13 and SF-20, corresponding to the monoclinic type (Table 1), whereas pyrrhotite from samples S-494 and S-643 is less sulfurous: $(Fe,Ni)_{0.89-0.91}S_{1.00}$ with Ni content in the range 0.45–0.67 wt.%.

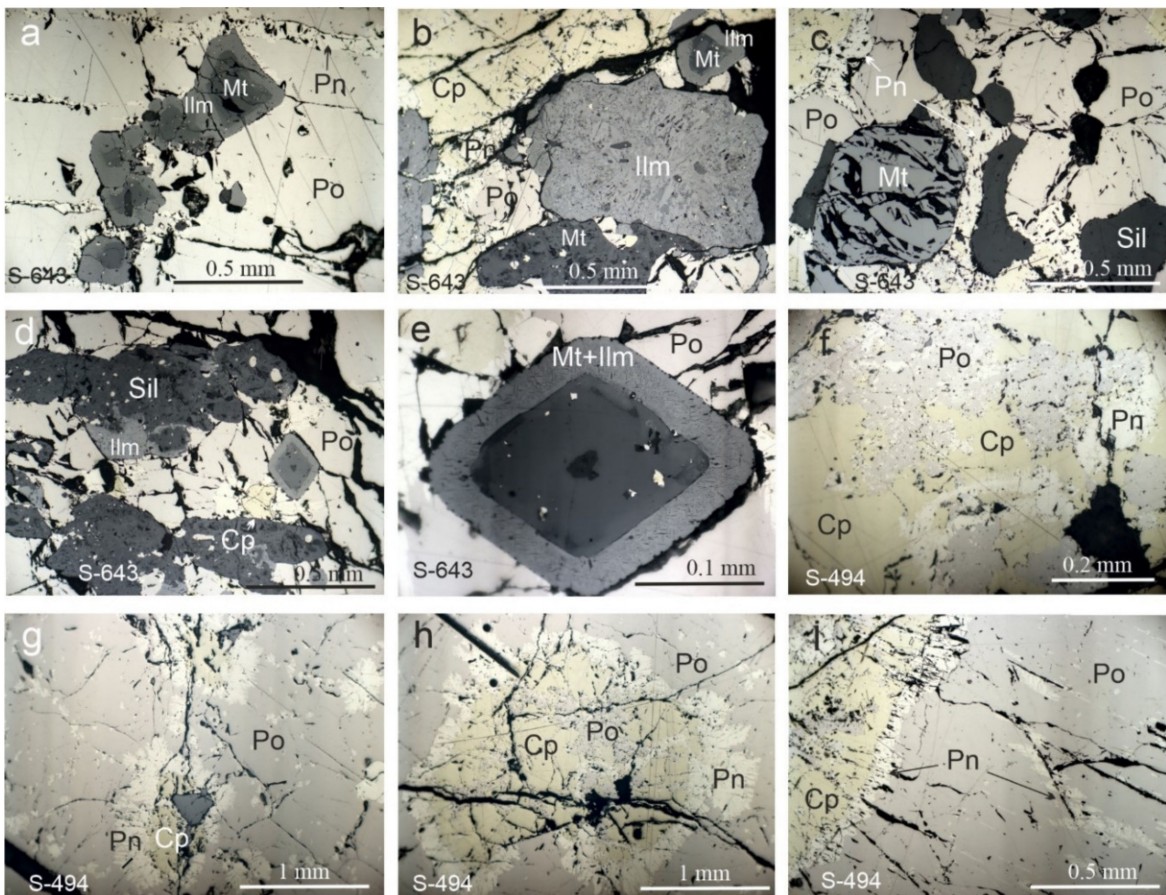

**Figure 4.** Textures and parageneses of sulfides in massive ores from boreholes S-643 (**a**–**e**) and S-494 (**f**–**i**), reflected light. (**a**) Magnetite (Mt)–ilmenite (Ilm) zoned grains together with pentlandite veinlets in pyrrhotite (Po); (**b**) ilmenite-grain-bearing sphene and perovskite lamellae and sulfide microinclusions; (**c**) droplet-shape magnetite grain in a sulfide matrix composed of pyrrhotite (Po), pentlandite (Pn), and chalcopyrite (Cp); (**d**–**e**) crystals of ulvospinel with apatite in core, surrounded by a zone of magnetite–ilmenite (Mt + Ilm) exsolutions; (**f**–**i**) chalcopyrite segregations surrounded by a rim of pentlandite, in pyrrhotite.

These compositions correspond to the hexagonal variety $Me_8S_9$ and $Me_9S_{10}$. The Ni content in pentlandite prevails over the Fe content in samples from SF-13 and SF-20 boreholes while in some samples from S-643 there are Fe-rich varieties (Tables 1 and 2), both of which contain ubiquitous Co between 0.75–2.11 wt.%. Chalcopyrite is stoichiometric and does not present discrete variations. Additionally, there is an unnamed phase, $Fe_3NiS_4$, occurring as inclusions in sperrylite, which corresponds to Fe-SS, see composition in Table 1 and Figure 5a. Pyrite occurs only in samples from the SF-13 borehole, in association with the most S-rich monoclinic pyrrhotite. Ni is a minor element in pyrite and varies around 0.52–0.68 wt.%. The compositions of all sulfides are summarized and shown in Figure 5a. Pyrite also contains variable amounts of Co, which are considered low on the EDS elemental mapping (Figure 6).

**Table 1.** Compositions of sulfides from samples SF-13, SF-20, S-494, and S-643 (wt.%).

| No. | Sample | Fe | Co | Ni | Cu | S | Pb | Total | Formula | Mineral |
|-----|--------|-----|------|-------|-------|-------|------|--------|---------|---------|
| 1 | SF-13 | 58.95 | | | | 39.39 | 0.27 | 98.62 | $Fe_{0.86}S_{1.00}$ | Pyrrhotite $Fe_{1-x}S$ |
| 2 | SF-13 | 58.98 | | | | 39.44 | 0.24 | 98.69 | $Fe_{0.86}S_{1.00}$ | Pyrrhotite $Fe_{1-x}S$ |
| 3 | SF-13 | 58.76 | | 0.58 | | 39.39 | 0.29 | 99.02 | $(Fe_{0.86}Ni_{0.01})_{0.87}S_{1.00}$ | Pyrrhotite $Fe_{1-x}S$ |
| 4 | SF-13 | 58.23 | | 0.59 | | 39.43 | 0.28 | 98.53 | $(Fe_{0.85}Ni_{0.01})_{0.86}S_{1.00}$ | Pyrrhotite $Fe_{1-x}S$ |
| 5 | SF-20 | 59.76 | | 0.52 | | 39.79 | | 100.07 | $(Fe_{0.86}Ni_{0.01})_{0.87}S_{1.00}$ | Pyrrhotite $Fe_{1-x}S$ |
| 6 | SF-20 | 59.47 | | | | 39.19 | | 98.66 | $Fe_{0.87}S_{1.00}$ | Pyrrhotite $Fe_{1-x}S$ |
| 7 | S-494 | 60.43 | | 0.54 | | 38.60 | | 99.57 | $(Fe_{0.90}Ni_{0.01})_{0.91}S_{1.00}$ | Pyrrhotite $Fe_{1-x}S$ |
| 8 | S-494 | 60.37 | | 0.64 | | 38.83 | | 99.84 | $(Fe_{0.89}Ni_{0.01})_{0.90}S_{1.00}$ | Pyrrhotite $Fe_{1-x}S$ |
| 9 | S-494 | 59.96 | | 0.64 | | 38.78 | | 99.38 | $(Fe_{0.89}Ni_{0.01})_{0.90}S_{1.00}$ | Pyrrhotite $Fe_{1-x}S$ |
| 10 | S-494 | 60.18 | | 0.67 | | 38.58 | | 99.42 | $(Fe_{0.90}Ni_{0.01})_{0.91}S_{1.00}$ | Pyrrhotite $Fe_{1-x}S$ |
| 11 | S-643 | 59.33 | | 0.59 | | 38.65 | | 98.57 | $(Fe_{0.88}Ni_{0.01})_{0.89}S_{1.00}$ | Pyrrhotite $Fe_{1-x}S$ |
| 12 | S-643 | 60.02 | | 0.45 | | 38.53 | | 99.00 | $(Fe_{0.89}Ni_{0.01})_{0.90}S_{1.00}$ | Pyrrhotite $Fe_{1-x}S$ |
| 13 | SF-13 | 28.88 | 0.75 | 37.66 | | 32.61 | | 99.89 | $(Ni_{4.98}Fe_{4.02}Co_{0.10})_{9.10}S_{7.90}$ | Pentlandite $(Fe,Ni)_9S_8$ |
| 14 | SF-13 | 30.62 | 1.04 | 34.49 | | 33.36 | | 99.51 | $(Ni_{4.55}Fe_{4.25}Co_{0.14})_{8.94}S_{8.06}$ | Pentlandite $(Fe,Ni)_9S_8$ |
| 15 | SF-13 | 29.41 | 1.07 | 35.81 | | 33.38 | | 99.67 | $(Ni_{4.72}Fe_{4.08}Co_{0.14})_{8.94}S_{8.06}$ | Pentlandite $(Fe,Ni)_9S_8$ |
| 16 | SF-20 | 30.49 | 1.36 | 34.68 | | 33.85 | | 100.38 | $(Ni_{4.53}Fe_{4.19}Co_{0.18})_{8.90}S_{8.10}$ | Pentlandite $(Fe,Ni)_9S_8$ |
| 17 | SF-20 | 30.65 | 1.48 | 34.75 | | 33.48 | | 100.36 | $(Ni_{4.55}Fe_{4.22}Co_{0.19})_{8.96}S_{8.03}$ | Pentlandite $(Fe,Ni)_9S_8$ |
| 18 | SF-20 | 30.23 | 1.49 | 35.36 | | 33.73 | | 100.81 | $(Ni_{4.61}Fe_{4.14}Co_{0.19})_{8.94}S_{8.05}$ | Pentlandite $(Fe,Ni)_9S_8$ |
| 19 | SF-20 | 30.65 | 1.32 | 34.01 | | 33.39 | | 99.37 | $(Ni_{4.49}Fe_{4.26}Co_{0.17})_{8.92}S_{8.08}$ | Pentlandite $(Fe,Ni)_9S_8$ |
| 20 | S-494 | 30.17 | 1.24 | 35.17 | | 32.84 | | 99.42 | $(Ni_{4.66}Fe_{4.20}Co_{0.16})_{9.02}S_{7.97}$ | Pentlandite $(Fe,Ni)_9S_8$ |
| 21 | S-494 | 30.28 | 1.29 | 35.28 | | 32.92 | | 99.77 | $(Ni_{4.66}Fe_{4.20}Co_{0.17})_{9.03}S_{7.96}$ | Pentlandite $(Fe,Ni)_9S_8$ |
| 22 | S-494 | 30.19 | 1.33 | 35.42 | | 32.94 | | 99.88 | $(Ni_{4.67}Fe_{4.19}Co_{0.18})_{9.04}S_{7.96}$ | Pentlandite $(Fe,Ni)_9S_8$ |
| 23 | S-494 | 30.10 | 1.38 | 35.32 | | 32.97 | | 99.77 | $(Ni_{4.66}Fe_{4.18}Co_{0.18})_{9.02}S_{7.97}$ | Pentlandite $(Fe,Ni)_9S_8$ |
| 24 | S-643 | 31.30 | 1.47 | 33.41 | | 32.77 | | 98.95 | $(Ni_{4.45}Fe_{4.38}Co_{0.19})_{9.02}S_{7.98}$ | Pentlandite $(Fe,Ni)_9S_8$ |
| 25 | S-643 | 31.57 | 1.35 | 32.35 | | 32.73 | | 98.00 | $(Fe_{4.45}Ni_{4.34}Co_{0.18})_{8.97}S_{8.03}$ | Pentlandite $(Fe,Ni)_9S_8$ |
| 26 | S-643 | 31.01 | 1.78 | 32.72 | | 32.90 | | 98.41 | $(Ni_{4.37}Fe_{4.35}Co_{0.24})_{8.96}S_{8.04}$ | Pentlandite $(Fe,Ni)_9S_8$ |
| 27 | S-643 | 31.03 | 2.11 | 32.49 | | 32.76 | | 98.39 | $(Fe_{4.36}Ni_{4.34}Co_{0.28})_{8.98}S_{8.02}$ | Pentlandite $(Fe,Ni)_9S_8$ |
| 28 | S-643 | 31.30 | 1.47 | 33.41 | | 32.77 | | 98.95 | $(Ni_{4.45}Fe_{4.38}Co_{0.19})_{9.02}S_{7.98}$ | Pentlandite $(Fe,Ni)_9S_8$ |
| 29 | SF-13 | 30.32 | | | 34.80 | 35.01 | | 100.13 | $Cu_{1.00}Fe_{0.99}S_{2.00}$ | Chalcopyrite $CuFeS_2$ |
| 30 | SF-13 | 29.62 | | | 34.77 | 34.00 | | 98.39 | $Cu_{1.02}Fe_{0.99}S_{1.98}$ | Chalcopyrite $CuFeS_2$ |
| 31 | SF-13 | 29.36 | | | 34.81 | 34.34 | | 98.51 | $Cu_{1.02}Fe_{0.98}S_{2.00}$ | Chalcopyrite $CuFeS_2$ |
| 32 | SF-20 | 30.18 | | | 33.54 | 35.14 | | 98.86 | $Cu_{0.98}Fe_{1.00}S_{2.03}$ | Chalcopyrite $CuFeS_2$ |
| 33 | S-494 | 30.47 | | | 34.12 | 34.82 | | 99.41 | $Cu_{0.99}Fe_{1.01}S_{2.00}$ | Chalcopyrite $CuFeS_2$ |
| 34 | S-494 | 30.74 | | | 33.80 | 34.48 | | 99.02 | $Cu_{0.99}Fe_{1.02}S_{1.99}$ | Chalcopyrite $CuFeS_2$ |
| 35 | S-494 | 30.87 | | | 33.68 | 34.83 | | 99.38 | $Cu_{0.98}Fe_{1.02}S_{2.00}$ | Chalcopyrite $CuFeS_2$ |
| 36 | S-494 | 30.73 | | | 33.95 | 34.53 | | 99.20 | $Cu_{0.99}Fe_{1.02}S_{1.99}$ | Chalcopyrite $CuFeS_2$ |
| 37 | S-643 | 29.57 | | | 33.08 | 35.12 | | 98.43 | $Cu_{0.97}Fe_{0.98}S_{2.01}$ | Chalcopyrite $CuFeS_2$ |
| 38 | SF-13 | 46.38 | | | | 53.44 | | 99.82 | $Fe_{1.00}S_{2.00}$ | Pyrite $FeS_2$ |
| 39 | SF-13 | 43.96 | 0.50 | 0.17 | | 53.45 | | 98.08 | $(Fe_{0.97}Co_{0.01})_{0.98}S_{2.02}$ | Pyrite $FeS_2$ |
| 40 | SF-13 | 44.48 | 1.91 | 0.06 | | 53.48 | | 99.93 | $(Fe_{0.96}Co_{0.04})_{1.00}S_{2.00}$ | Pyrite $FeS_2$ |
| 41 | SF-13 | 44.22 | 2.09 | 0.07 | | 53.30 | | 99.68 | $(Fe_{0.95}Co_{0.04})_{0.99}S_{2.00}$ | Pyrite $FeS_2$ |
| 42 | SF-13 | 43.73 | 2.48 | 0.07 | | 52.61 | | 98.89 | $(Fe_{0.95}Co_{0.05})_{1.00}S_{1.99}$ | Pyrite $FeS_2$ |
| 43 | S-494 | 45.95 | 0.81 | 17.56 | | 35.64 | | 99.96 | $(Fe_{2.93}Co_{0.05})_{2.98}Ni_{1.07}S_{3.96}$ | Unnamed $Fe_3NiS_4$ |

Note. SF-13 and S-494—WDS data; SF-20 and S-643—EDS data.

The EDS elemental maps were obtained from a specific area of a polished sample mount (SF-13) selected because within this site Co in pyrite appears unevenly distributed. This fragment represents the marginal zone of segregation of chalcopyrite (Figure 6a). All sulfides from this fragment were analyzed by an EDS microanalyzer (see Materials and Methods). The numeration of points in Figure 6b corresponds to the No. column in Table 2. The resulting elemental maps show that pyrite presents a heterogeneous distribution and is characterized by oscillatory zoning. Cobalt-rich zones are found in the core, mid-part, and on the rim of the grain (Figure 6c,e). The compositions of pyrite in Table 2 reflect this zoning. The maximum Co content in this grain is 2.91 wt.% whereas in the next grain is even higher (3.98 wt.%). Pentlandite in the fragment studied occupies an interstitial position between pyrite and pyrrhotite, as evidenced by the distribution of Ni in Figure 7d.

The distribution of copper in Figure 7f demonstrates that pyrite does not go beyond the chalcopyrite segregation, but is rather restricted to its margins.

**Table 2.** Compositions of sulfides from samples SF-13 (wt.%) shown in Figure 6.

| No. | Fe | Co | Ni | Cu | S | Total | Formula |
|---|---|---|---|---|---|---|---|
| 1 | 30.39 | | | 33.64 | 34.55 | 98.58 | $Fe_{1.01}Cu_{0.98}S_{2.00}$ |
| 2 | 31.02 | | | 33.81 | 34.61 | 99.44 | $Fe_{1.03}Cu_{0.98}S_{1.99}$ |
| 3 | 30.82 | | | 33.37 | 34.87 | 99.06 | $Fe_{1.02}Cu_{0.97}S_{2.01}$ |
| 4 | 30.70 | | | 33.33 | 34.50 | 98.53 | $Fe_{1.02}Cu_{0.98}S_{2.00}$ |
| 5 | 29.36 | 1.12 | 36.37 | | 33.58 | 100.43 | $(Ni_{4.76}Fe_{4.04}Co_{0.15})_{8.95}S_{8.05}$ |
| 6 | 29.38 | 1.16 | 36.07 | | 33.20 | 99.81 | $(Ni_{4.76}Fe_{4.07}Co_{0.15})_{8.98}S_{8.02}$ |
| 7 | 28.99 | 0.96 | 36.31 | 0.44 | 33.22 | 99.92 | $(Ni_{4.79}Fe_{4.02}Co_{0.13}Cu_{0.05})_{8.99}S_{8.02}$ |
| 8 | 29.09 | 1.31 | 36.39 | | 33.14 | 99.93 | $(Ni_{4.80}Fe_{4.03}Co_{0.17})_{9.00}S_{8.00}$ |
| 9 | 59.57 | | 0.51 | | 39.24 | 99.75 | $(Fe_{0.87}Ni_{0.01})_{0.88}S_{1.00}$ |
| 10 | 59.52 | | 0.68 | | 39.37 | 98.81 | $(Fe_{0.87}Ni_{0.01})_{0.88}S_{1.00}$ |
| 11 | 44.15 | 2.35 | | | 52.68 | 99.18 | $(Fe_{0.96}Co_{0.05})_{1.01}S_{1.99}$ |
| 12 | 44.31 | 2.58 | | | 53.01 | 99.90 | $(Fe_{0.96}Co_{0.05})_{1.01}S_{1.99}$ |
| 13 | 46.47 | 0.14 | | | 52.96 | 99.57 | $(Fe_{1.00}Co_{0.01})_{1.01}S_{1.99}$ |
| 14 | 46.27 | 0.65 | | | 52.98 | 99.9 | $(Fe_{1.00}Co_{0.01})S_{1.99}$ |
| 15 | 44.77 | 2.01 | | | 53.08 | 99.86 | $(Fe_{0.97}Co_{0.04})_{1.01}S_{1.99}$ |
| 16 | 45.4 | 1.17 | | | 53.10 | 99.67 | $(Fe_{0.98}Co_{0.02})_{1.00}S_{2.00}$ |
| 17 | 44.10 | 2.91 | | | 53.36 | 100.37 | $(Fe_{0.95}Co_{0.06})_{1.01}S_{1.99}$ |
| 18 | 42.77 | 3.98 | | | 52.79 | 99.54 | $(Fe_{0.93}Co_{0.08})_{1.01}S_{1.99}$ |

Note. EDS data. The analysis numbers (No.) correspond to the points in Figure 7b.

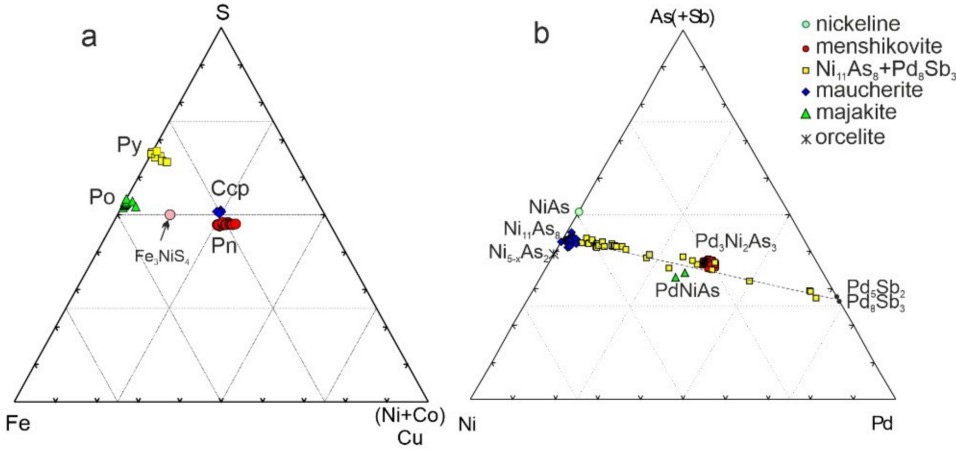

**Figure 5.** Compositions of sulfides (**a**) and minerals of the Pd–Ni–As(Sb) system (**b**) from the Skalisty mine.

### 3.3. Morphology and Compositions of PGM

Numerous PGM grains and Au–Ag alloys were found within samples from boreholes S-494 and S-643 in heavy concentrates of crushed samples, while none were observed in massive ores from boreholes SF-13 and SF-20, which intersect the core of the main orebody and where maximum PGE Contents were determined. Given the high contents of PGE in these ores, including Rh, Ru, and Ir, the possible presence of PGE solid solutions in base metal sulfides (BMS) is suggested.

Among the identified PGM, sperrylite $PtAs_2$ prevails over the rest and occurs as euhedral grains 30–60 μm in size (Figures 7a and 8b). Other Pt minerals present in these boreholes are isoferroplatinum and cooperite (Pt,Pd,Ni)S represented by 20–40 μm grains (Figure 8a,d). All the Pt minerals, and sometimes menshikovite $Pd_3Ni_2As_3$ (Figure 8c), are found as discrete grains, whereas the majority of the Pd minerals: menshikovite, majakite PdNiAs, sobolevskite Pd(Bi,Sb), stibiopalladinite $Pd_5(Sb,As)_2$, mertieite II $Pd_8(Sb,As)_2$, and native gold (smallest grains 3–5 μm) are found included in maucherite $Ni_{11}As_8$ grains

(Figures 7b–e and 8f–h). PGE arsenides are the most common among Pd minerals. Menshikovite and majakite are intergrown rarely with nickeline (NiAs) similarly. Mertieite II sometimes occurs as emulsion impregnations in maucherite, which implies presumably exsolution of the high-temperature solid solution Ni–Pd–As–Sb (Figure 5b). An unnamed phase of composition $(Pd,Ni)_6Bi_4Sb$ was identified—three inclusions in maucherite (Figure 8h). Menshikovite grains vary in size from small inclusions to subhedral grains intergrown with maucherite (Figure 8e,f), and as discrete large grains (Figure 8c) exceeding 100 µm. Sperrylite sometimes pervades the droplet-like sulfide inclusions (Figure 8a).

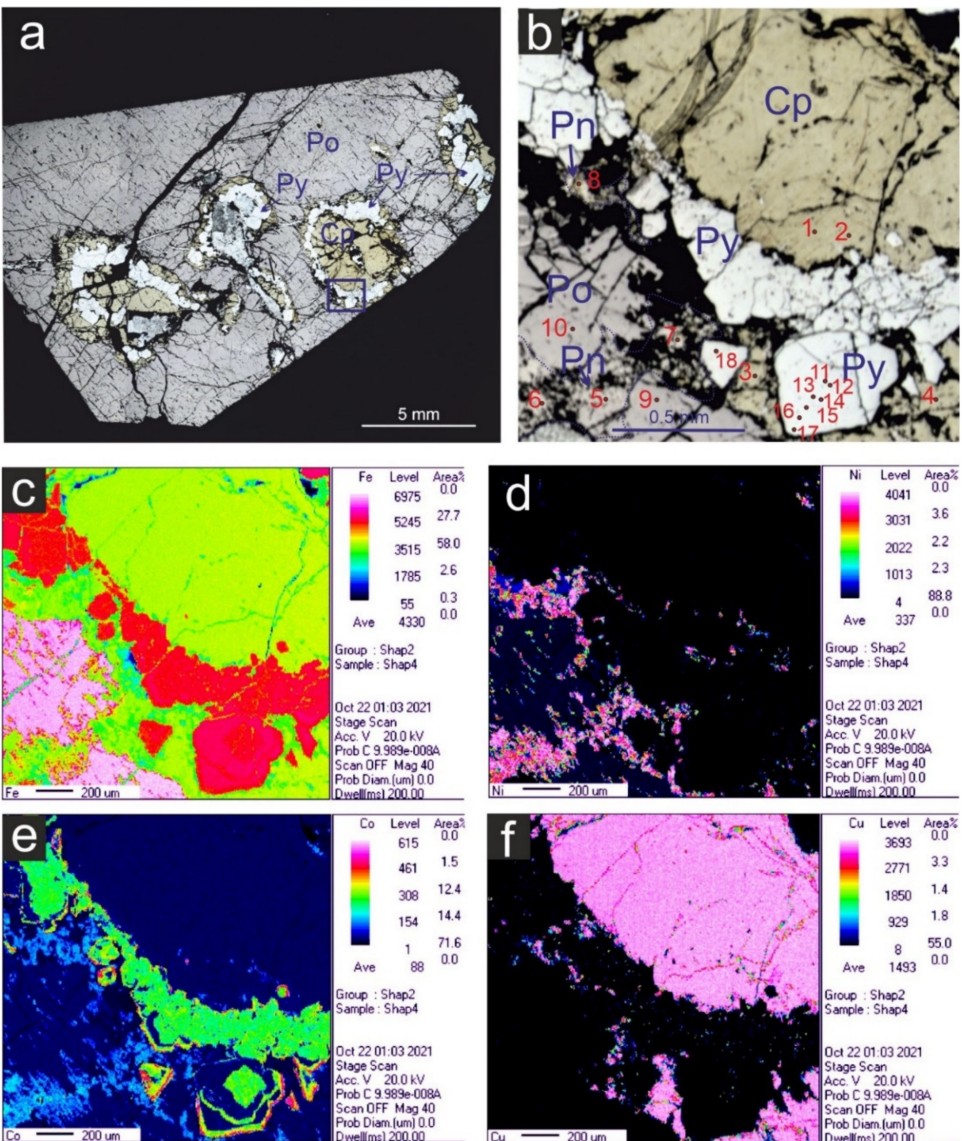

**Figure 6.** Polished sample mount of massive ores from borehole SF-13 (**a**), site of study within this sample—selected area for elemental maps (**b**), and EDS elemental distribution maps of the designated site of study: Fe (**c**), Ni (**d**), Co (**e**), and Cu (**f**). Points and numbers in (**b**) correspond to the analysis number shown in Table 2. Abbreviations: Py—pyrite, Po—pyrrhotite, Cp—chalcopyrite, Pn—pentlandite.

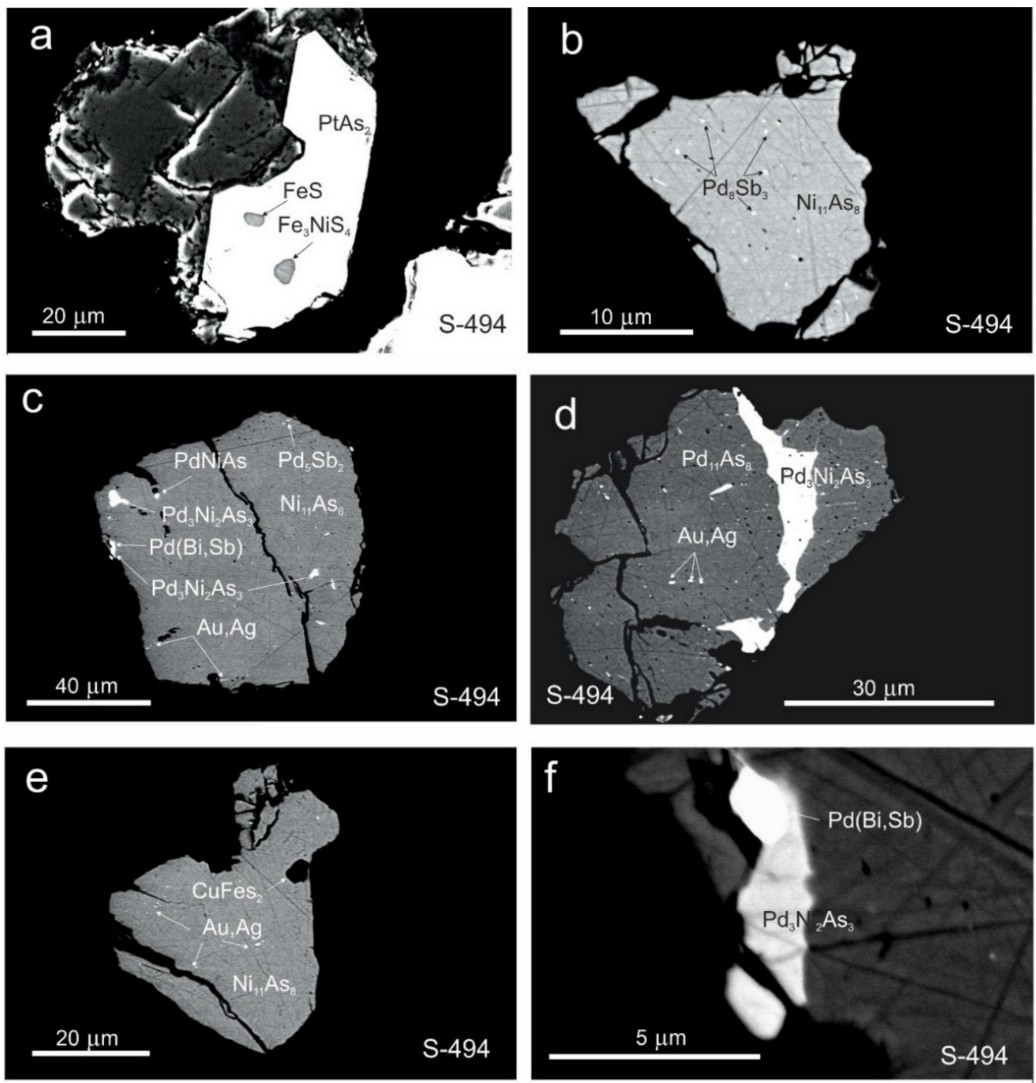

**Figure 7.** Morphology and microparageneses of PGM from samples of borehole S-494, SEM image: (**a**) crystals of sperrylite $PtAs_2$ with droplet-shaped inclusions of pyrrhotite FeS and Fe-SS ($Fe_3NiS_4$); (**b–e**) maucherite grains with tiny inclusions of mertieite II $Pd_8As_3$ (**b**), menshikovite $Pd_3Ni_2As_3$, majakite PdNiAs, stibiopalladinite $Pd_5Sb_2$, sobolevskite Pd (Bi,Sb), and gold (Au,Ag) (**c–e**); (**f**) grains of maucherite intergrown with menshikovite and sobolevskite.

Maucherite contains Pd as a minor element in the range 1.61–4.31 wt.% (Table 3). These Pd contents were obtained in large homogeneous grains. Another ubiquitous minor element in maucherite is Co, which occurs in the range 0.21–0.89 wt.%. An intermediate composition between menshikovite and majakite has also been established (Figure 5b). Nickeline is a much less common phase, but may also contain Pd up to 0.38 wt.%. Other PGMs are less diverse in composition: cooperite, sobolevskite, sperrylite, mertieite II, stibiopalladinite, and isoferroplatinum. Sobolevskite contains about 20 mol.% of sudburyite. Mertieite II is characterized by a constant composition with an approximate formula $Pd_8Sb_{2.5}As_{0.5}$. Isoferroplatinum is stoichiometric; cooperite contains Pd and Ni, less than 1 wt.% each (Table 4). Stibiopalladinite inclusions are too small to be quantified as well as the Au–Ag alloys. Several analyses for gold–silver alloys were obtained by removing matrix elements (Ni, As) and recalculating to 100 at.%. As a result, it was established that the compositions of Au–Ag alloys vary widely from 38 to 100 at.%, Au showing the most distinctive fineness 55–77 at.% (Figure 9).

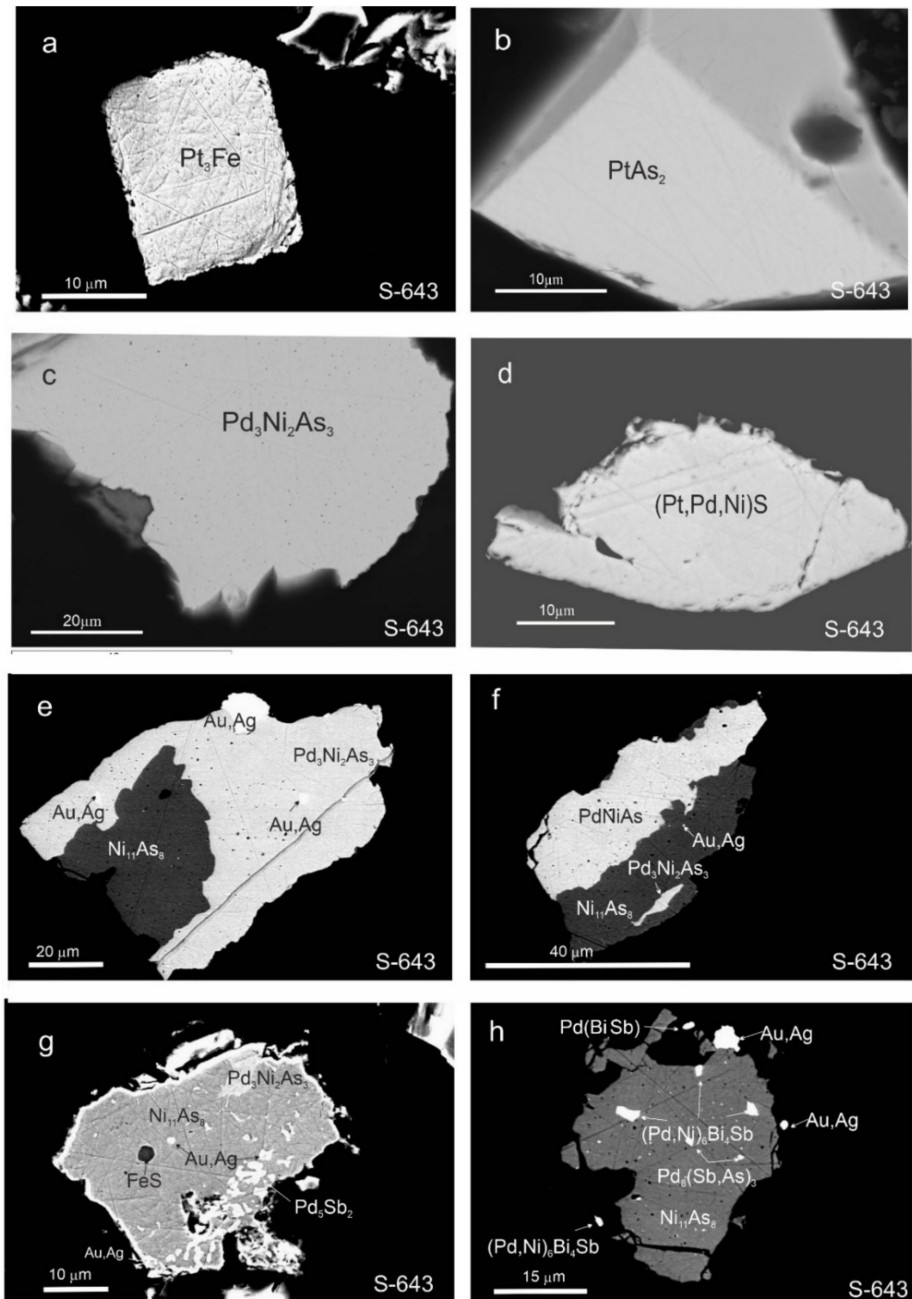

**Figure 8.** SEM image showing the morphology and microparageneses of PGM, sample S-643: (**a**) iso-ferroplatinum grain; (**b**) sperrylite crystal; (**c**) menshikovite grain; (**d**) cooperate grain; (**e**,**f**) intergrowths of menshikovite $Pd_3Ni_2As_3$ and maucherite $Ni_{11}As_8$; (**g**,**h**) inclusions of PGM, sulfide, and gold (Au,Ag) in maucherite.

### 3.4. Geochemical Features of the Massive Ores from the Skalisty Mine

  The singular geochemical characteristics of the massive ores of the Skalisty mine according to Norilskgeologiya LLC (internal report) were presented earlier in [30,31] and in this study, shown in Tables 5 and 6. The high contents of PGEs (14–32 ppm), a significant proportion of which are IPGE and Rh have been found exclusively in massive exocontact ores of the Skalisty mine (Figure 10), which may represent a typomorphic feature. Additionally, these ores are nickel-rich (Ni content ranging 4.8–5.95 wt.%, and Ni/Cu ratios varying from 1.3 to 1.9. The total Contents of PGE vary in a range 14–32 ppm; palladium prevails over platinum, Pd/Pt ratios are 7–8 in massive chalcopyrite–pyrrhotite ores of the S-643 and S-494 boreholes, while these ratios range from 4 to 4.5 in SF-13 and

SF-20 boreholes. The high IPGE contents—Ru up to 3.84 ppm and Ir up to 1.10 ppm, and Rh up to 7.31 ppm (Table 6)—are higher than the previously described by [30]. The Contents of Ir + Ru + Rh in 100% sulfide varies in the range 5.3–15.5 ppm (SF-13, SF-20, and S-494), and do not exceed 1 ppm in the massive ore of the S-643 borehole. The ratios (Pd + Pt)/(Ir + Ru + Rh) or PPGE/IPGE + Rh in samples of massive ores present low values for most samples (1–2), and it corresponds to 37 for the S-643 borehole.

**Table 3.** Composition of Ni and Pd arsenides of the Skalisty mine (wt.%).

| No. | Sample | Co | Ni | As | Pd | Total | Formula | Mineral |
|---|---|---|---|---|---|---|---|---|
| 1 | S-643 | 0.56 | 43.57 | 56.56 | 0.00 | 100.69 | $(Ni_{0.99}Co_{0.01})_{1.00}As_{1.00}$ | Nickeline NiAs |
| 2 | S-643 | 0.50 | 43.24 | 56.04 | 0.38 | 100.16 | $(Ni_{0.98}Co_{0.01})_{0.99}As_{1.00}$ | Nickeline NiAs |
| 3 | S-643 | 0.77 | 49.65 | 47.62 | 1.61 | 99.65 | $(Ni_{10.65}Pd_{0.19}Co_{0.16})_{11.00}As_{8.00}$ | Maucherite $Ni_{11}As_8$ |
| 4 | S-643 | 0.83 | 49.51 | 47.71 | 1.66 | 99.71 | $(Ni_{10.61}Pd_{0.20}Co_{0.18})_{10.99}As_{8.01}$ | Maucherite $Ni_{11}As_8$ |
| 5 | S-643 | 0.56 | 48.67 | 47.19 | 2.58 | 99.00 | $(Ni_{10.55}Pd_{0.31}Co_{0.12})_{10.98}As_{8.02}$ | Maucherite $Ni_{11}As_8$ |
| 6 | S-643 | | 50.27 | 47.68 | 1.82 | 99.77 | $(Ni_{10.78}Pd_{0.22})_{10.98}As_{8.01}$ | Maucherite $Ni_{11}As_8$ |
| 7 | S-643 | 0.21 | 48.29 | 46.97 | 4.31 | 99.78 | $(Ni_{10.47}Pd_{0.52}Co_{0.05})_{11.04}As_{7.97}$ | Maucherite $Ni_{11}As_8$ |
| 8 | S-643 | 0.89 | 49.08 | 47.61 | 2.67 | 100.25 | $(Ni_{10.51}Pd_{0.32}Co_{0.19})_{11.02}As_{7.99}$ | Maucherite $Ni_{11}As_8$ |
| 9 | S-643 | 0.28 | 49.80 | 48.13 | 2.49 | 100.70 | $(Ni_{10.61}Pd_{0.29}Co_{0.06})_{10.96}As_{8.03}$ | Maucherite $Ni_{11}As_8$ |
| 10 | S-643 | 0.34 | 49.35 | 47.34 | 2.21 | 99.24 | $(Ni_{10.66}Pd_{0.26}Co_{0.07})_{10.99}As_{8.01}$ | Maucherite $Ni_{11}As_8$ |
| 11 | S-643 | 0.46 | 49.66 | 47.88 | 2.52 | 100.52 | $(Ni_{10.60}Pd_{0.30}Co_{0.10})_{11.00}As_{8.01}$ | Maucherite $Ni_{11}As_8$ |
| 12 | S-643 | 0.45 | 49.12 | 47.37 | 2.72 | 99.66 | $(Ni_{10.58}Pd_{0.32}Co_{0.10})_{11.00}As_{8.00}$ | Maucherite $Ni_{11}As_8$ |
| 13 | S-643 | 0.29 | 48.24 | 46.08 | 2.49 | 97.10 | $(Ni_{10.66}Pd_{0.30}Co_{0.06})_{11.02}As_{7.98}$ | Maucherite $Ni_{11}As_8$ |
| 14 | S-643 | 0.58 | 48.71 | 47.19 | 2.96 | 99.44 | $(Ni_{10.53}Pd_{0.35}Co_{0.12})_{11.00}As_{7.99}$ | Maucherite $Ni_{11}As_8$ |
| 15 | S-643 | 0.42 | 49.48 | 47.35 | 2.82 | 100.07 | $(Ni_{10.62}Pd_{0.33}Co_{0.09})_{11.04}As_{7.96}$ | Maucherite $Ni_{11}As_8$ |
| 16 | S-643 | | 18.01 | 32.78 | 48.51 | 99.30 | $Pd_{3.04}Ni_{2.05}As_{2.92}$ | Menshikovite $Pd_3Ni_2As_3$ |
| 17 | S-643 | | 18.11 | 33.11 | 48.44 | 99.66 | $Pd_{3.02}Ni_{2.05}As_{2.93}$ | Menshikovite $Pd_3Ni_2As_3$ |
| 18 | S-643 | | 17.93 | 32.85 | 48.28 | 99.06 | $Pd_{3.03}Ni_{2.04}As_{2.93}$ | Menshikovite $Pd_3Ni_2As_3$ |
| 19 | S-643 | | 18.13 | 33.09 | 48.80 | 100.02 | $Pd_{3.03}Ni_{2.04}As_{2.92}$ | Menshikovite $Pd_3Ni_2As_3$ |
| 20 | S-643 | | 18.27 | 33.12 | 48.67 | 100.06 | $Pd_{3.02}Ni_{2.06}As_{2.92}$ | Menshikovite $Pd_3Ni_2As_3$ |
| 21 | S-643 | | 18.15 | 33.09 | 48.56 | 99.80 | $Pd_{3.02}Ni_{2.05}As_{2.93}$ | Menshikovite $Pd_3Ni_2As_3$ |
| 22 | S-643 | | 18.76 | 32.93 | 48.07 | 99.76 | $Pd_{2.98}Ni_{2.11}As_{2.90}$ | Menshikovite $Pd_3Ni_2As_3$ |
| 23 | S-643 | | 24.63 | 31.83 | 44.95 | 101.41 | $Pd_{1.00}Ni_{0.99}As_{1.01}$ | Majakite PdNiAs |
| 24 | S-643 | | 24.26 | 31.89 | 44.31 | 100.46 | $Pd_{1.00}Ni_{0.99}As_{1.02}$ | Majakite PdNiAs |

Note. All analyses are EDS data.

**Table 4.** Composition of PGM of the Skalisty mine (wt.%).

| No. | Sample | Fe | Ni | Pt | Pd | Bi | Sb | As | S | Te | Total | Formula | Mineral |
|---|---|---|---|---|---|---|---|---|---|---|---|---|---|
| 1 | S-494 | | 0.55 | 85.17 | 0.60 | | | | 14.33 | | 100.65 | $(Pt_{0.97}Ni_{0.02}Pd_{0.01})_{1.00}S_{0.99}$ | Cooperite PtS |
| 2 | S-494 | | 0.96 | 83.87 | 1.00 | | | | 14.62 | | 100.45 | $(Pt_{0.94}Ni_{0.04}Pd_{0.02})_{1.00}S_{1.00}$ | Cooperite PtS |
| 3 | S-494 | | 1.62 | 82.51 | 1.44 | | | | 14.46 | | 100.03 | $(Pt_{0.92}Ni_{0.06}Pd_{0.03})_{1.01}S_{0.99}$ | Cooperite PtS |
| 4 | S494 | | | | 34.64 | 62.88 | 1.27 | | | | 98.79 | $Pd_{1.02}(Bi_{0.94}Sb_{0.03})_{0.97}$ | Sobolevskite Pd(Bi,Sb) |
| 5 | S647 | | | | 36.28 | 55.85 | 7.61 | | | | 99.74 | $Pd_{1.02}(Bi_{0.80}Sb_{0.19})_{0.99}$ | Sobolevskite Pd(Bi,Sb) |
| 6 | S648 | | | | 36.18 | 54.46 | 7.26 | | | 1.41 | 99.31 | $Pd_{1.01}(Bi_{0.78}Sb_{0.18}Te_{0.03})_{0.99}$ | Sobolevskite Pd(Bi,Sb) |
| 7 | S649 | | | | 36.46 | 54.15 | 7.85 | | | 1.57 | 100.03 | $Pd_{1.01}(Bi_{0.76}Sb_{0.19}Te_{0.04})_{0.99}$ | Sobolevskite Pd(Bi,Sb) |
| 8 | S643 | | | 55.31 | | | | 42.85 | | | 98.16 | $Pt_{0.99}As_{2.01}$ | Sperrylite $PtAs_2$ |
| 9 | S494 | | | 56.16 | | | | 42.32 | 0.27 | | 99.13* | $(Pt_{1.00}Rh_{0.01})_{1.01}(As_{1.96}S_{0.03})_{1.99}$ | sperrylite $PtAs_2$ |
| 10 | S494 | | | 56.44 | | | | 42.13 | 0.21 | | 98.78 | $Pt_{1.01}(As_{1.97}S_{0.02})_{1.99}$ | Sperrylite $PtAs_2$ |
| 11 | S494 | | | 56.34 | | | | 42.09 | 0.23 | | 98.66 | $Pt_{1.01}(As_{1.96}S_{0.03})_{1.99}$ | Sperrylite $PtAs_2$ |
| 12 | S643 | | | | 71.54 | | 25.40 | 3.01 | | | 99.95 | $Pd_{8.03}(Sb_{2.49}As_{0.48})_{2.97}$ | Mertieite II $Pd_8(Sb,As)_3$ |
| 13 | S643 | | | | 71.12 | | 25.74 | 3.24 | | | 100.10 | $Pd_{7.97}(Sb_{2.52}As_{0.52})_{3.04}$ | Mertieite II $Pd_8(Sb,As)_3$ |
| 14 | S643 | | | | 71.29 | | 25.59 | 3.07 | | | 99.95 | $Pd_{8.00}(Sb_{2.51}As_{0.49})_{3.00}$ | Mertieite II $Pd_8(Sb,As)_3$ |
| 15 | S643 | 9.54 | | 88.91 | | | | | | | 98.45 | $Pt_{2.91}Fe_{1.09}$ | Isoferroplatinum $Pt_3Fe$ |
| 16 | S643 | | 2.90 | | 36.28 | 54.05 | 7.61 | | | | 100.84 | $(Pd_{5.27}Ni_{0.76})_{6.03}Bi_{4.00}Sb_{0.97}$ | Unnamed $(Pd,Ni)_6Bi_4Sb$ |

Note. All analyses are EDS data.

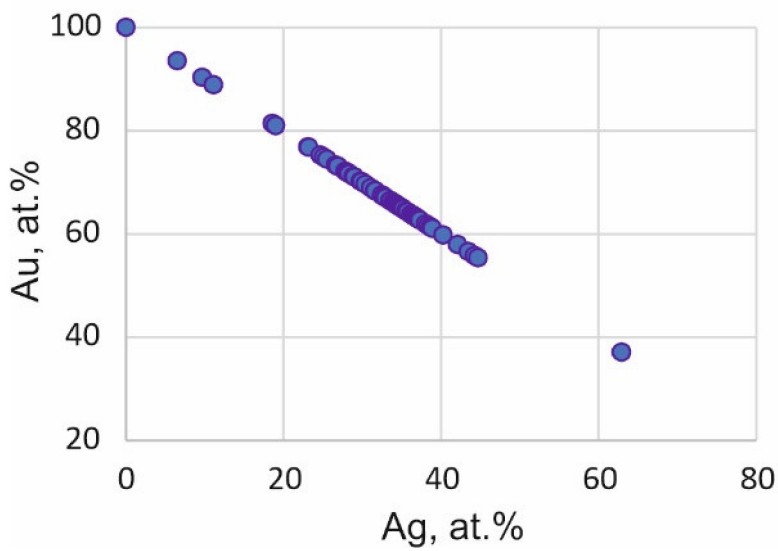

**Figure 9.** Compositions of Au–Ag alloys (at.%).

**Table 5.** Content of elements in ores and the Content of these elements, recalculated to 100% sulfide, from the Skalisty mine. Northeastern part of the Talnakh intrusion.

| No. | Sample | Depth, m | Pt | Pd | Au | Ir | Rh | Ru | Co | Ni | Cu | S |
|---|---|---|---|---|---|---|---|---|---|---|---|---|
| | | | | | ppm | | | | | | wt.% | |
| 1 | SF-13 | 908.8 | 2.37 | 10.77 | 0.16 | 1.10 | 7.31 | 3.84 | 0.18 | 4.96 | 2.57 | 36.05 |
| 2 | SF-20 | 919.7 | 3.23 | 13.26 | 0.30 | 0.70 | 5.22 | 2.68 | 0.17 | 4.81 | 3.49 | 36.00 |
| 3 | SF-20 | 921.0 | 2.90 | 13.00 | 0.31 | 0.95 | 6.79 | 3.61 | 0.18 | 4.96 | 3.87 | 36.00 |
| 4 | S-494 | 0.0 | 1.32 | 9.40 | 0.08 | 0.42 | 3.70 | 1.07 | 0.22 | 5.95 | 3.48 | 36.90 |
| 5 | S-643 | 2.7 | 1.36 | 11.30 | 0.05 | 0.01 | 0.36 | 0.03 | 0.19 | 4.94 | 2.60 | 35.30 |
| | | | | | In 100% sulfide | | | | | | | |
| No | Sample | m | Pt | Pd | Au | Ir | Rh | Ru | Co | Ni | Cu | S |
| 1 | SF-13 | 908.8 | 3.01 | 13.66 | 0.20 | 1.40 | 9.27 | 4.87 | 0.22 | 6.29 | 3.26 | 45.73 |
| 2 | SF-20 | 919.7 | 3.39 | 13.91 | 0.31 | 0.73 | 5.48 | 2.81 | 0.17 | 5.04 | 3.66 | 37.74 |
| 3 | SF-20 | 921.0 | 3.03 | 13.61 | 0.33 | 0.99 | 7.11 | 3.77 | 0.18 | 5.19 | 4.05 | 37.68 |
| 4 | S-494 | 0.0 | 1.35 | 9.58 | 0.08 | 0.43 | 3.77 | 1.09 | 0.22 | 6.06 | 3.55 | 37.61 |
| 5 | S-643 | 2.7 | 1.46 | 12.09 | 0.05 | 0.01 | 0.39 | 0.04 | 0.20 | 5.29 | 2.78 | 37.78 |

**Table 6.** Geochemical features of the massive exocontact ores (in 100% sulfide) from the Skalisty mine. Northeastern part of the Talnakh intrusion.

| No. | Sample | Depth, m | Ni/Cu | Pd/Pt | PGE | PGE/S | Ir + Ru + Rh | PPGE/IPGE | Cu/Pd | Rh/Cu | Pd/Au |
|---|---|---|---|---|---|---|---|---|---|---|---|
| 1 | SF-13 | 908.8 | 1.93 | 4.54 | 32.22 | $705 \times 10^{-7}$ | 15.55 | 1.1 | 2386 | $2844 \times 10^{-7}$ | 68.2 |
| 2 | SF-20 | 919.7 | 1.38 | 4.11 | 26.31 | $697 \times 10^{-7}$ | 9.02 | 1.9 | 2631 | $1497 \times 10^{-7}$ | 44.7 |
| 3 | SF-20 | 921.0 | 1.28 | 4.48 | 28.52 | $757 \times 10^{-7}$ | 11.87 | 1.4 | 2976 | $1755 \times 10^{-7}$ | 41.7 |
| 4 | S-494 | 0.0 | 1.71 | 7.12 | 16.21 | $431 \times 10^{-7}$ | 5.29 | 2.1 | 3702 | $1063 \times 10^{-7}$ | 117.5 |
| 5 | S-643 | 2.7 | 1.90 | 8.31 | 13.98 | $370 \times 10^{-7}$ | 0.43 | 31.2 | 2301 | $138 \times 10^{-7}$ | 226.0 |

The Content of chalcophile elements in massive ores is higher than chondrite C1 in the SF-13 and SF-20 samples (Figure 11); the pattern has a sawtooth-like configuration due to a maximum in Rh and Pd and a minimum in Au. Massive ores from sample S-643 differ from the other samples due to lower IPGE contents that cause a negative anomaly in the IPGE area, resulting from a higher degree of fractionation.

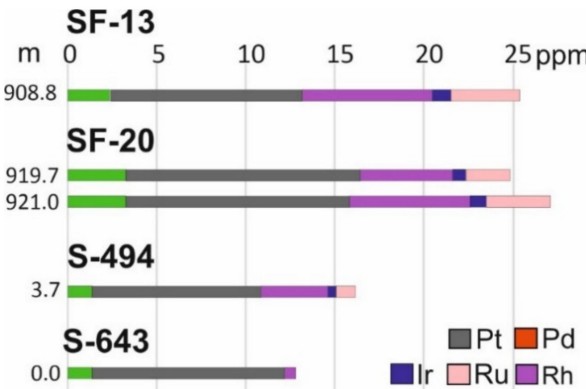

**Figure 10.** PGE Content (ppm) in the ores of the studied boreholes from the Skalisty mine, Talnakh intrusion.

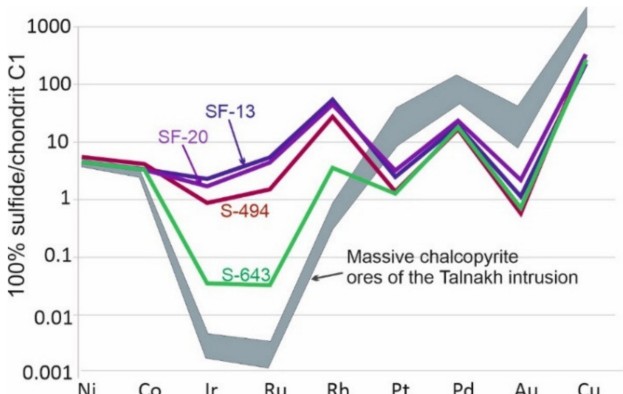

**Figure 11.** Plots of 100% sulfide/chondrite C1-normalized [40] average tenors of chalcophile elements in massive ores. Gray area—massive chalcopyrite ore of the Talnakh intrusion, Southern-2 orebody (Yz-5 and Yuz-12 samples) [21].

A brief comparison on the distributions of chalcophile elements in the pyrrhotite ores against massive chalcopyrite ores of the southwestern branch of the Talnakh intrusion [18] clearly shows a substantial difference in the degree of fractionation; the ores from southwestern Talnakh are much more fractioned according to their lower IPGE and higher Cu content.

### 3.5. Distribution of Chalcophile Elements among the Base Metal Sulfides

For LA–ICP–MS studies, two samples with different degrees of fractionation were selected: (a) pyrrhotite ores (or Cu-poor) containing Po, $Pn_1$ and $Pn_2$, Cp, and Py (SF-13), in which discrete PGMs were not found; and (b) S-494 chalcopyrite–pyrrhotite (Cu-bearing), containing Po, $Pn_1$, Cp, and minerals of Pd (Pt)–Ni–As–(Sb) systems. The LA–ICP–MS study of both samples shows that almost all Pd is concentrated in pentlandite of two generations: granular ($Pn_1$) and lamellas ($Pn_2$) (Figure 12). The average values are comparable in SF-13 and S-494, but in more fractionated ores, its content is rather higher—240.5 ppm (Table 7). A small proportion of Pd is also found in pyrrhotite (up to 3.48 ppm), but only in the Cu-poor ore sample.

In pyrrhotite ores, where Pt minerals have not been found, this element is concentrated in Pn of both generations and Cp. It is below detection limit in Po (Figure 12); however, greater amounts of Pt are found in Py (up to 62.56 ppm).

LA-ICP-MS studies have identified minerals that concentrate Rh and IPGE (Rh + It + Ru up to 11.87 ppm in whole rock). Rh, Ru, Ir, and Os are preferentially concentrated in Py, the sum of which may average up to 28 ppm (Figure 12), with maximum values of these contents: Rh—36.00, Ru—38.26, Ir—6.89, and Os—4.77 (ppm). The remaining portion of

these elements (except for Os) is almost evenly distributed between Po (up to 6.64 ppm) and Pn$_2$ (up to 9.50 ppm). In Cp-bearing sample S-494, Rh and Ir are concentrated in Pn (up to 3.17 and 0.44 ppm) and Po (up to 3.93 and 0.42 ppm), respectively, whereas Ru and Os are below the detection limit (Table 7).

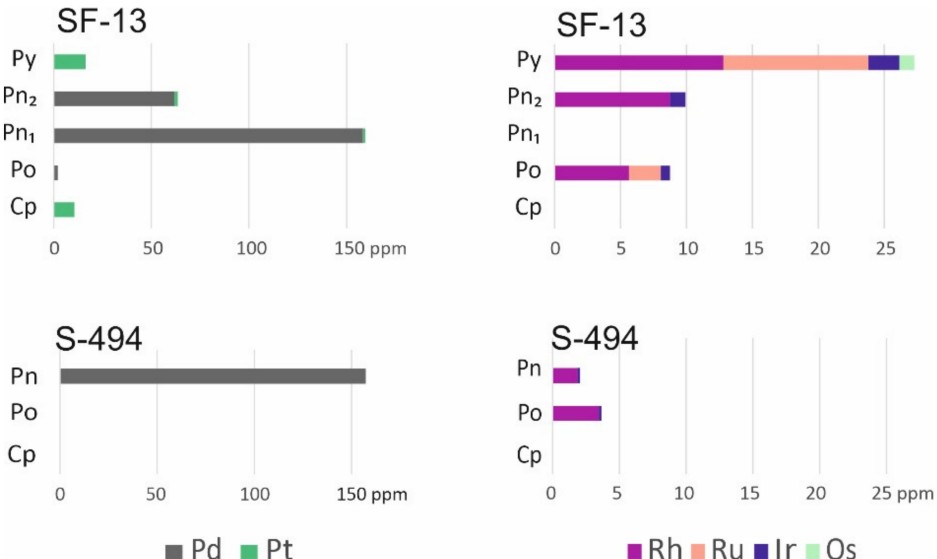

**Figure 12.** PGE content in BMS for the selected samples: SF-13—pyrrhotite ores (Cu-poor) and S-494—chalcopyrite–pyrrhotite (Cu-bearing), average values in the mineral, presented in Table 6.

Trace contents of As are present in all BMS of both samples (Table 7), for most cases in the amount of tens ppm; nonetheless, the As content reaches hundreds of ppm in Cp, Pn$_2$, and Py in Cu-poor ores, where discrete grains of Pd and Ni arsenides were not found. Nickel as a trace element is enriched in Po and Py, whereas Co concentrates mostly in Fe-rich BMS, which is consistent with EDS and WDS sulfide analyses.

In addition, it should be noted that the maximum Rh contents in pyrite coincide with the Co-enriched zones shown in Figure 7. Moreover, weak positive correlation between the Co and IPGE + Rh grades in pyrrhotite and pentlandite is evident (Figure 13).

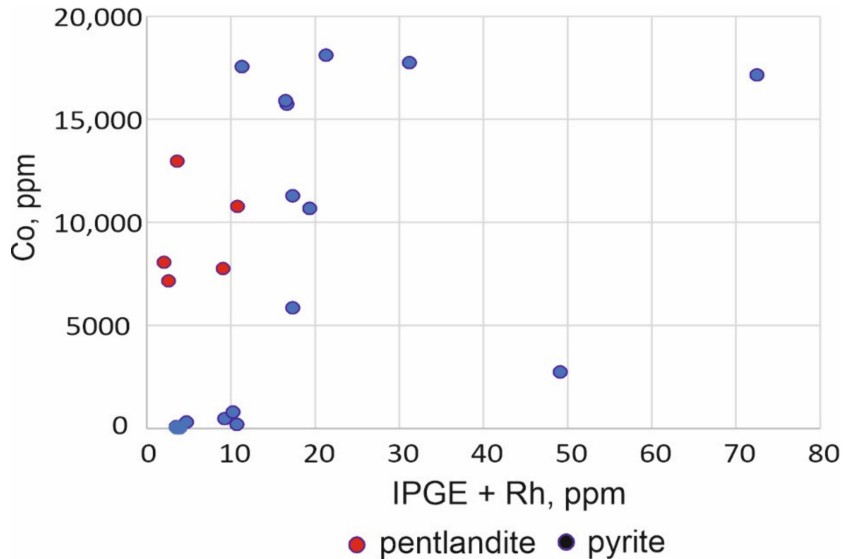

**Figure 13.** A total sum of IPGE + Rh versus Co contents (ppm) in pentlandite and pyrite, LA-ICP-MS data. The graph shows a positive correlation for both types of sulfides.

**Table 7.** Trace element contents in sulfides, as determined by LA–ICP–MS.

| Sample # | Mineral | | D, µm | $^{59}$Co ppm | $^{60}$Ni ppm | $^{65}$Cu ppm | $^{66}$Zn ppm | $^{75}$As ppm | $^{77}$Se ppm | $^{101}$Ru ppm | $^{103}$Rh ppm | $^{105}$Pd ppm | $^{107}$Ag ppm | $^{185}$Re ppm | $^{189}$Os ppm | $^{191}$Ir ppm | $^{195}$Pt ppm | $^{206}$Pb ppm | $^{209}$Bi ppm |
|---|---|---|---|---|---|---|---|---|---|---|---|---|---|---|---|---|---|---|---|
| S-494 | ccp | line | 40 | - | - | n.a. | 195.9 | 186.2 | - | - | - | - | - | - | - | - | - | - | - |
| | SD | | | | | | 28.64 | 17.67 | | | | | | | | | | | |
| S-494 | po | spot | 60 | 37.32 | 5074 | - | - | 71.9 | 67.14 | - | 3.36 | - | - | - | - | 0.27 | - | 3.09 | - |
| | SD | | | 2.38 | 279.3 | | | 7.13 | 17.23 | | 0.35 | | | | | 0.08 | | 0.43 | |
| S-494 | po | spot | 60 | 85.11 | 6664 | | | 75.45 | 61.57 | - | 3.1 | | | | | 0.42 | - | 1.99 | - |
| | SD | | | 15.09 | 585.5 | | | 7.24 | 13.28 | | 0.39 | | | | | 0.11 | | 0.3 | |
| S-494 | po | line | 40 | 27.31 | 5541 | | | 39.28 | - | - | 3.93 | - | | | | - | - | 2.31 | - |
| | SD | | | 3.81 | 574.1 | | | 10.27 | | | 0.85 | | | | | | | 0.57 | |
| S-494 | pn | spot | 60 | 12.25 | n.a. | 73.75 | - | 191.1 | - | - | - | 162.8 | - | - | - | - | - | 9.39 | - |
| | SD | | | 421.4 | | 11.44 | | 13.55 | | | | 9.63 | | | | | | 1.35 | |
| S-494 | pn | spot | 60 | 7164 | n.a. | - | - | 105 | 42.44 | - | 2.31 | 101.7 | 2.38 | - | - | 0.29 | - | 2.18 | - |
| | SD | | | 662.2 | | | | 9.33 | 12.34 | | 0.28 | 9.03 | 0.45 | | | 0.09 | | 0.28 | |
| S-494 | pn | line | 40 | 12.97 | n.a. | 38.15 | - | 190.8 | 40.29 | - | 3.17 | 240.5 | 1.25 | - | - | 0.44 | - | 2.41 | - |
| | SD | | | 457.8 | | 12.51 | | 10.38 | 11.6 | | 0.34 | 11.6 | 0.31 | | | 0.13 | | 0.31 | |
| S-494 | pn | line | 40 | 8067 | n.a. | - | - | 95.72 | 69.23 | - | 2.08 | 124.6 | 4.24 | - | - | - | - | 2.08 | - |
| | SD | | | 1084 | | | | 18.96 | 33.11 | | 0.66 | 21.67 | 1.17 | | | | | 0.54 | |
| SF-13 | ccp | line | 40 | - | 45.46 | n.a. | 478.9 | 177 | - | - | - | - | 9.91 | - | - | - | 21.22 | 52.74 | - |
| | SD | | | | 6.97 | | 96.99 | 19.09 | | | | | 1.91 | | | | 3.33 | 6.36 | |
| SF-13 | po with pn2 | spot | 60 | 789.1 | 25.15 | - | - | 98.13 | 67.57 | 2.82 | 6.64 | 3.47 | 2.18 | - | - | 0.79 | - | 1.44 | - |
| | SD | | | 54.06 | 1645 | | | 8.23 | 15.28 | 0.82 | 0.71 | 0.94 | 0.46 | | | 0.15 | | 0.29 | |
| SF-13 | po | spot | 60 | 311.8 | 9185 | - | - | 43.93 | 21.2 | 1.23 | 3.16 | 1.47 | - | 0.22 | - | 0.35 | - | 0.31 | - |
| | SD | | | 23.04 | 675.8 | | | 3.69 | 6.76 | 0.34 | 0.31 | 0.46 | | 0.07 | | 0.07 | | 0.07 | |
| SF-13 | po | spot | 60 | 205.1 | 9234 | - | - | 88.24 | 57.27 | 3.62 | 6.25 | - | - | - | - | 0.82 | - | 0.74 | - |
| | SD | | | 28.05 | 993.5 | | | 8.18 | 15.19 | 0.88 | 0.58 | | | | | 0.18 | | 0.18 | |
| SF-13 | pn | line | 40 | 9199 | n.a. | 61.53 | - | 99.09 | - | - | - | 167.4 | - | - | - | - | - | 28.69 | - |
| | SD | | | 976.1 | | 21.59 | | 19.52 | | | | 21.89 | | | | | | 7.1 | |
| SF-13 | pn | line | 40 | 5739 | n.a. | 6005 | 52.06 | 24.26 | - | - | - | 149.4 | 1.57 | - | - | - | 2.4 | 11.8 | - |
| | SD | | | 1035 | | 1745 | 13.31 | 6.51 | | | | 23.66 | 0.89 | | | | 0.5 | 2.69 | |
| SF-13 | pn2 | spot | 60 | 10.76 | n.a. | - | - | 188.7 | 95.55 | - | 9.5 | 81.39 | 20.05 | - | - | 1.31 | 1.69 | 21.35 | - |
| | SD | | | 489.5 | | | | 15.33 | 28.9 | | 0.88 | 8.26 | 2.42 | | | 0.29 | 0.5 | 1.83 | |
| SF-13 | py core | spot | 60 | 17.55 | 356.5 | 66.16 | - | 92.72 | 75.62 | 4.77 | 4.74 | - | - | 0.75 | 0.62 | 1.21 | 7.29 | 1.16 | 0.32 |
| | SD | | | 810.2 | 14.4 | 10.8 | | 5.85 | 9 | 0.59 | 0.32 | | | 0.1 | 0.13 | 0.1 | 0.63 | 0.13 | 0.05 |
| SF-13 | py rim | spot | 60 | 156.8 | 11.69 | - | - | 94.15 | 78.38 | - | - | - | - | - | - | - | - | - | - |
| | SD | | | 9.28 | 1.44 | | | 5.57 | 9.74 | | | | | | | | | | |
| SF-13 | py core | line | 40 | 17.15 | 470.8 | - | - | 121.5 | - | 38.26 | 22.59 | - | - | 2.97 | 4.77 | 6.89 | 19.85 | 7.29 | 4.46 |
| | SD | | | 1395 | 35.56 | | | 11.7 | | 9.45 | 2.93 | | | 0.77 | 1.31 | 0.9 | 2.75 | 1.53 | 0.81 |
| SF-13 | py core | line | 40 | 2746 | 136.8 | - | - | 117.5 | 123.3 | 10.8 | 36 | - | - | - | - | 2.34 | 62.56 | 4.59 | - |
| | SD | | | 495.1 | 32.86 | | | 17.55 | 39.61 | 3.92 | 10.35 | | | | | 0.81 | 13.95 | 1.67 | |
| SF-13 | py rim | line | 40 | 241.2 | 83.02 | - | - | 131.3 | 80.24 | - | - | - | - | - | - | - | - | - | - |
| | SD | | | 60.29 | 12.99 | | | 10.2 | 19.94 | | | | | | | | | | |
| SF-13 | py | spot | 60 | 15726 | 548.2 | 480.2 | - | 48.46 | 78.25 | 6.14 | 8.58 | - | - | 0.85 | 0.63 | 1.3 | 6.34 | 3.51 | 0.47 |
| | SD | | | 320.1 | 17.34 | 84.47 | | 3.42 | 9.78 | 0.76 | 0.71 | | | 0.11 | 0.12 | 0.12 | 0.36 | 0.26 | 0.05 |
| SF-13 | py | line | 40 | 17740 | 646.4 | 2179 | - | 44.06 | 70.69 | 13.43 | 13.5 | - | 0.72 | 1.47 | 1.21 | 3.09 | 12.58 | 8.31 | 2.65 |
| | SD | | | 489.1 | 32.46 | 489.1 | | 3.11 | 12.45 | 2.62 | 2 | | 0.2 | 0.25 | 0.28 | 0.36 | 0.98 | 0.84 | 0.24 |
| | Detection limit | | | 0.75 | 1.58 | 3.15 | 1.53 | 2.29 | 2.67 | 0.01 | 0.02 | 0.05 | 0.1 | 0.038 | 0.004 | 0.003 | 0.003 | 0.074 | 0.053 |

Note. SD—standard deviation of measurements, "-"—below detection limits, n.a.—not analyzed, since they are the main elements that compose the mineral.

## 4. Discussion

### 4.1. Degree of Fractionation of the Rh-IPGE Bearing Massive Sulfide Ores

Variations in Rh content relative to Cu, which are often used as an index of fractional crystallization [1], show that the massive ores from the Skalisty mine belong to MSS, i.e., initial sulfide melt that underwent very weak fractionation, less than 10% (Figure 14). The Rh/Cu ratios in these ores are extremely high, and correspond to values of $1850 \times 10^{-7}$ at very high Contents of IPGE + Rh (15.55 ppm in 100% sulfide). Contrastingly, the ores are located on the northwestern flank of the main ore body (sample S-643), which also represent cumulus of MSS with a higher degree of fractionation (less than 60%) compared to the previous ores. The Rh/Cu ratios are situated above the line $100 \times 10^{-7}$ for these samples (Figure 14), and correspond to values of roughly $140 \times 10^{-7}$ (Table 6); they are expressed by a steeper positive slope pattern of elements (Figure 11) due to the low content of IPGE in 100% sulfide (0.43 ppm). The difference in the degree of fractionation of massive ores from the Skalisty mine is also revealed by the Pd/Pt ratio, which ratio in Cu-poor ores is 4.1 whereas in Cu-bearing ores equals 8.3.

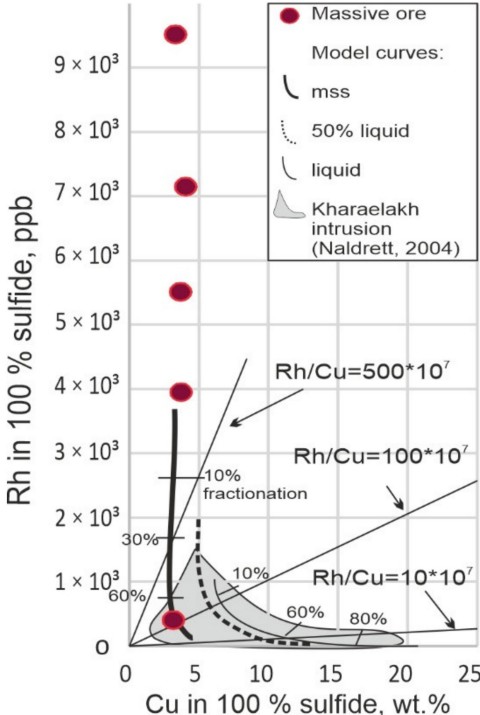

**Figure 14.** Variations in Rh content relative to Cu for different samples of the Skalisty mine compared with data from the Norilsk region concerning model curves of Rayleigh fractionation [1].

Weakly fractionated ores formed from primitive melts are generally enriched in nickel. Ni/Cu ratios in the massive pyrrhotite ores of the Skalisty mine (1.3–1.9) are in a broad sense comparable with other massive pyrrhotite ores of mafic and ultramafic complexes (Ni/Cu is 1.2–1.4) [12,23,41,42], a feature that distinguishes them from the Cu-rich mooihoekite ores of the Oktyabr'sky deposit (Ni/Cu is 0.11), the chalcopyrite ores of the Southern-2 orebody of the Talnakh intrusion (0.2) [21], and the disseminated ores in taxitic gabbro–dolerite of the Norilsk 1 intrusion, in which the Ni/Cu ratio is 0.5–0.8 on average [20]. Analogous or slightly higher Ni/Cu values to those of the Skalisty mine are also typical for disseminated ores of the picritic gabbro–dolerites of the Norilsk deposits, for the Jinchuan (China), Kabanga (Tanzania), and Pechenga (Russia) deposits [43].

### 4.2. Composition of Pyrrhotite in the Massive Sulfide Ores

Pyrrhotite is the most abundant mineral among sulfide ores of the Skalisty mine. As shown in the results, in weakly fractionated ores (SF-13 and SF-20) the monoclinic variety $Me_7S_8$ is the norm, whereas hexagonal pyrrhotite $Me_8S_9$–$Me_9S_{10}$ was mainly observed in more fractionated ores (S-494, S-643). Monoclinic pyrrhotite in association with pyrite, similar to the studied paragenesis in samples from borehole SF-13, crystallizes at higher temperatures and sulfur fugacity, compared to the hexagonal variety [44]. Thus, it is likely that the ores of the marginal part of the main orebody, which are more fractionated, crystallized at lower temperatures and lower conditions of sulfur fugacity. Moreover, Rh content up to 33.3 and 133 ppm was determined in pyrrhotite and pentlandite, respectively, from low-sulfide ores of Norilsk 1 intrusion [45].

### 4.3. Evolution of the Sulfide Melt and Mechanism of Pyrite Formation in Weakly Fractionated Ores

Grains of euhedral pyrite were found only in weakly fractionated ores (SF-13 and SF-20) in close association with Cp and minor Pn. It is known that MSS cumulates can trap a Cu-rich liquid during the formation of massive sulfides; in this case, peritectic Pn develops at the contact between the MSS and the incorporated liquid [29,46,47]; this is observed in poorly fractionated samples SF-13. A thin rim of pentlandite along chalcopyrite segregations (Figure 15), and pentlandite plates exsolved from MSS below 550 °C diffusing toward grain boundaries to form granular Pn, along with brush textures at <250 °C [48]. These brush textures are typical of samples C-494 and C-643 (Figure 4g,h). Sulfide liquid below 700 ± 25 °C, crystallizes to form an intermediate solid solution (ISS) [48] which exsolves into Ccp and Po at 350 °C if conditions of $fS_2$ are moderate, and into Ccp and Py at 500 °C in high-sulfur systems [34], which may correspond to the textures observed in Figure 15.

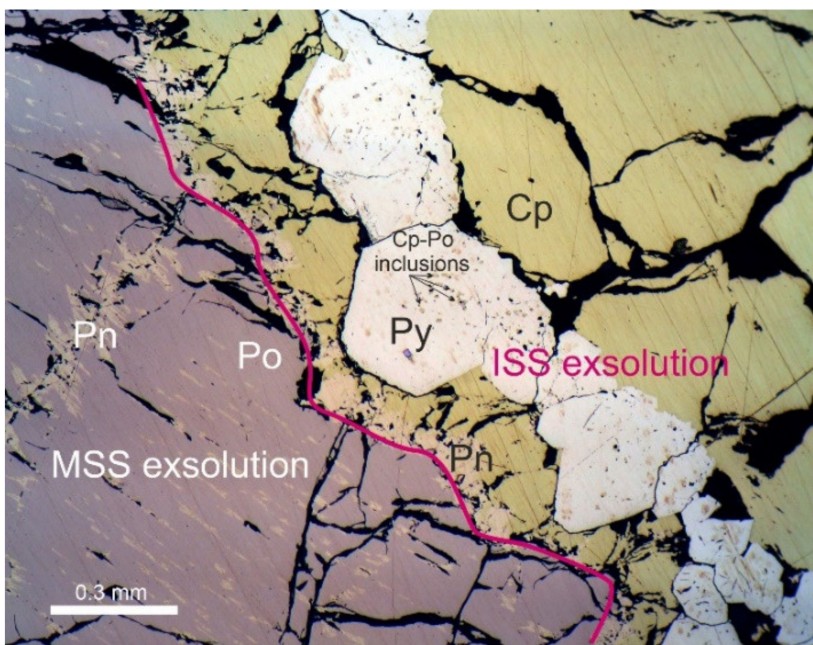

**Figure 15.** Contact between MSS (Po + Pn) and ISS (Cp + Py + Pn) exsolutions.

However, according to most experimental studies, pyrite is most likely to be exsolved from MSS below 750 °C [49–52]. As an alternative, pyrite can also be formed by the replacement of primary pyrrhotite [50], which has also been confirmed by natural findings [53–55]. Pyrite with complex zonation can be formed by a subsolidus reaction involving both MSS and ISS, and does not require hydrothermal processes [56]. Although the mechanisms of formation of the pyrite investigated here are still a matter of research, taking into account the observations on textural relationships in sample SF-13, consisting of granular pyrite in

Cp matrix, which does not go beyond the chalcopyrite segregations i.e., Py is restricted to the margins of Ccp grains (Figure 15).

### 4.4. PGE Enrichment of Sulfide Minerals

The Contents of Os (up to 4.8 ppm), Ir (up to 6.9 ppm), Ru (up to 38.3 ppm), Rh (up to 36.0 ppm), and Pt (up to 62.6 ppm) in studied pyrite are more significant compared to the contents of these elements in Po and Pn (Table 7). This enrichment in IPGE in BMS have not been previously observed nor reported in any type of massive sulfide ores of the Norilsk ore cluster, including the Cu-poor ones [29], whereas the pyrrhotites from sulfide droplets (Medvezhiy Creek mine), which were estimated to be enriched in IPGE up to 3.57 Os, 4.38 Ir, 13.9 Ir, and 90.9 Rh (ppm) [57]. However, similar IPGE Contents in pyrite (Ru up to 47 ppm, Os up to 7.8 ppm, and Ir up to 20 ppm) have also been described for the Main Sulfide Zone of the Great Zimbabwe Dike [58]. It is worth nothing that pyrites of non-magmatic origin may be enriched in IPGE as well, such as the hydrothermal Po in the Imandra layered complex, Kola Peninsula, containing Ru 10.96 wt.% [59] or the secondary Os-, Ir-, Ru-, Rh-containing Py formed after Po and Pn, from which these elements are inherited [60].

### 4.5. Partitioning of PGE among the Different Massive Sulfide Ores

Although the massive ores of the Skalisty mine (samples SF-13 and SF-20) are significantly enriched with PGE, including IPGE (up to 12.25 ppm), no discrete PGM grains have been found in these ores. On the contrary, Pt and Pd minerals (but not IPGE minerals) are present in ores from boreholes SF-643 and SF-494, which formed from a more fractionated sulfide melt. It was found that Pt minerals (cooperite, sperrylite, isoferroplatinum) and Pd minerals of the Pd(Ni)–As(Sb) systems occur in these samples (S-494 and S-643). If platinum is usually found as discrete minerals only, Pd, in addition to mineral phases, also may dissolve in pentlandite, where it typically ranges from ppb to ppm contents [61,62], and even sometimes reaches up to several wt.% [25,63,64]. Conversely, Os, Ir, Ru, and Rh are highly compatible with MSS and consequently with pyrrhotite and pentlandite, which exsolved from MSS [9,11,57,65,66]. The aforementioned is consistent with the absence of discrete IPGE-Rh bearing minerals in ores [67].

All the previous observations are also valid for the samples studied: both Po and Pn contain IPGE and Rh in the amount of 2–10 ppm (total) in more fractionated ores from the S-494 sample (Figure 12). However, IPGE Contents in BMS from the S-494 sample are depleted, while the Content of Pd, which is incompatible with MSS increased; a feature that was noticed earlier in [29]. Namely, the composition of BMS is partially controlled by the degree of fractional crystallization underwent by the sulfide liquid.

It should be noted, however, that non-solubility of noble metals in sulfides has been claimed recently by [68], instead suggesting their presence as nanoparticles. The Rh and IPGE clusters were initially present in the sulfide liquid before they entered the pentlandite lattice [69]. Nanoinclusions of Pt–Ir compounds in Rh-rich Pn, as in the Caridad chromite deposit of Cuba, are not considered to be exsolution products, but rather assumed to be formed in a silicate melt and later entrapped by a sulfide liquid [70], a hypothesis that is supported by the described mechanism of direct crystallization of PGM in primitive basaltic melt [71]. Our results, however, are inconsistent with these observations, since the selective capture of previously formed PGM clusters from silicate melts probably explain the formation of sulfides in systems undersaturated in sulfur, but not the formation of massive ores.

### 4.6. Conditions Controlling the Formation of PGM

In Cu-rich massive ores, palladium is fractionated into a residual sulfide liquid and then incorporated into nickeline or maucherite structures, and is then retained in these minerals even during serpentinization [72–74]. In Cu-poor pyrrhotite ores, Pd arsenides are formed as a result of exsolution from sulfides [18]. Exsolution textures in the Pd–Ni–As

(Sb, Bi) system are also observed in the studied samples; however, PGM exsolved from maucherite and not directly from BMS sulfides. Moreover, the amount of As in the sulfide melt was likely sufficient to segregate and separate the Sb(Bi)-bearing arsenide liquid from the sulfide melt before its crystallization. In this case, Pd and Ni could have been trapped by arsenide liquid, and after, under subsolidus conditions, the textures of decomposition of menshikovite, majakite, and mertieite II in maucherite were formed. The Pd Content in maucherite is homogeneous and varies in the range of 1.61–4.31 wt.%. According to experimental data [73] this Ni–As mineral with traces of Pd up to 5.5 wt.% is stable at temperatures above 450 °C.

All PGEs display a strong preference to the arsenide melt in accordance with the PGE partition coefficients between arsenide and sulfide melts ($K_D$ 20–2700.) According to [75–77], arsenic and other chalcogens in magmatic melts can form associations before reaching saturation in these elements. The separation (inmiscibility) of arsenide melts from sulfide liquid is an effective mechanism for the redistribution of PGE: the arsenide melt concentrates all PGEs if it separates before the crystallization of MSS; on the contrary, it exclusively concentrates elements that are incompatible with MSS e.g., Pd and Pt when liquation occurs after MSS crystallization [78]. Since exsolution texture of IPGE is completely absent in maucherite, the separation of the arsenide liquid had to occur after the crystallization of MSS in the here studied ores.

*4.7. The Problem of Forming IPGE-Rich Ores*

The observed elevated Contents of IPGE and Rh in the massive ores of the Skalisty mine and the absence of IPGE minerals allowed us to assume in a preliminary study [31] that the most probable location for these metals was in Po and Pn structure, according to the distribution coefficients between the MSS and the melt [9,11,64–66]. However, LA–ICP–MS studies carried out in this study on sulfides revealed an unexpected result: IPGE and Rh are unevenly distributed in pyrite. On the other hand, it is logical to assume that pyrite under study is magmatic in origin, as evidenced by its textural features. The presence of pyrite within chalcopyrite segregations reassures the possibility of its formation from ISS, with high sulfur activity. However, Rh remains in the sulfide melt only at low sulfur fugacity [79]. Another assumption is that the both MSS and ISS are possible sources for this pyrite, but the textural feature of the pyrite-band occurrence may indicate the reactional origin of pyrite between different ISS parts. Thus, the question on its anomalous enrichment in rhodium and IPGE remains open to debate if these elements are rather incompatible with ISS. Systematic and detailed LA–ISP–MS studies may provide an answer to this question in the near future.

Additionally, high Contents of IPGE in the sulfide melt should be the consequence of a magma initially enriched in these elements, which in a broader sense is related to high degrees of partial melting of the mantle. However, in models such as the conduit-type proposed by [1] for magmatic sulfide deposits, this is not strictly necessary because the sulfide melt can be enriched in IPGE from low-grade magma pulses passing through a staging magma chamber. According to [2], all the intrusions of the Norilsk-type are open, non-equilibrium magmatic systems, which provided a multiple supplies of magma into the chamber, as a result of which it is repeatedly saturated in all PGE. Directional crystallization experiments of sulfides + PGE [80] suggest the crystallization of MSS enriched in IPGE in the core of the intrusion (SF-13 and SF-20), and the "distillation" toward its frontal parts (apophyses) of more fractionated residual sulfide melts enriched in PPGE (S-494 and S-643). In poorly fractionated ores, all PGEs are dissolved in sulfides; in more fractionated ones, PPGE turns out to be partly forming discrete minerals and partly dissolved in sulfides, whereas IPGE and Rh all remain dissolved in sulfides

**5. Conclusions**

1.  Pyrrhotite (or Cu-poor) massive ores of the eastern flank of the Skalisty mine are unique in terms of their mineralogical and geochemical features. They are signifi-

cantly enriched in IPGE (with totals of Ir + Rh + Ru up to 12.25 ppm), unlike other ores of the Norilsk ore cluster. Samples from these ores show a variable degree of fractionation: from non-fractionated (SF-13) to weakly fractionated (SF-20), and further to increasingly fractionated S-494 and S-643. The last two are characterized by Ni speciation and high As and Sb potential in the ore-forming system: maucherite, menshikovite, majakite, and mertieite II—the most common PGM.

2. By increasing the degree of fractionation of the sulfide melt, the IPGE Content decreases, while the Pd Content increases. In the non-fractionated ores (SF-13 and SF-20), discrete PGM grains are absent, and all PGE (significant ore grades) are concentrated (dissolved) only in sulfides, while in more fractionated ores PGE may be in the form of discrete PGMs and (or) as solid solutions in sulfides.

3. The most important IPGE concentrator in the massive pyrrhotite ores of the Skalisty mine is pyrite, which appears only in the unfractionated SF-13 ores, and it contains up to 4.77 ppm Os, 6.89 ppm Ir, 38.26 ppm Ru, 36 ppm Rh, and 62.56 ppm Pt. This pyrite does not show any signs of hydrothermal reworking and is thought to have a magmatic origin.

4. High IPGE Contents in the sulfide melt can be due to several reasons: (1) a high degree of partial melting of the mantle, (2) enrichment in IPGE due to impulses of low-grade magma, or (3) in situ fractionation of the sulfide melt, i.e., in the magma chamber, and subsequent removal of a more fractionated residual melt to the periphery, favoring IPGE enrichment in MSS.

**Author Contributions:** Conceptualization, methodology, and writing—review and editing, validation, N.T.; sampling and general geology in the field, V.R.; LA-ICP-MS study and writing—review, V.B. and V.A.; preparation and description of samples, M.S.; English editing, J.G. All authors have read and agreed to the published version of the manuscript.

**Funding:** The studies were carried out within the framework of the state assignment of the IGM SB RAS and the framework of the state assignment of the IGEM RAS financed by the Ministry of Science and Higher Education of the Russian Federation. The research was carried out with the financial support of the project of the Russian Federation represented by the Ministry of Science and Higher Education of the Russian Federation No. 13.1902.21.0018 (agreement 075–15-2020–802).

**Acknowledgments:** We thank analysts M. Khlestov and V. Korolyuk for carrying out analytical procedures and providing quantitative EMF and WDS analyses, and N. Belkina and E. Shvetsova for technical assistance in preparing the manuscript. We are grateful to M. Yudovskaya for supporting our analytical LA–ICP–MS research.

**Conflicts of Interest:** The authors declare no conflict of interest.

## Appendix A

**Table A1.** Results of the RSM measurements for estimation of the accuracy and precision of the LA-ICP-MS analysis.

| Isotope | $^{59}$Co | $^{60}$Ni | $^{65}$Cu | $^{66}$Zn | $^{75}$As | $^{77}$Se | $^{107}$Ag | $^{191}$Ir | $^{195}$Pt | $^{206}$Pb | $^{209}$Bi |
|---|---|---|---|---|---|---|---|---|---|---|---|
| | ppm | ppm | ppm | ppm | ppm | ppm | ppm | ppm | ppm | ppm | ppm |
| Value [1] | 636 | 25,000 | 23,000 | 275 | 1050 | 310 | 155 | 60 | 50 | 90 | 120 |
| Value [2] | 60 | 97 | 134,000 | 210,000 | 65 | 51 | 50 | 63 | 52 | 68 | 60 |
| Average value *n* = 9 | 65.9 | 112.7 | 154,089 | 187,489 | 59.28 | 68.3 | 50.71 | 62.87 | 55.22 | 79.36 | 72.53 |
| SD | 5.8 | 11.91 | 14,171 | 24,429 | 9.4 | 8.58 | 2.57 | 5.43 | 7.13 | 4.16 | 3.24 |
| RSD.% | 8.79 | 10.56 | 9.2 | 13.03 | 13.76 | 16.92 | 4.03 | 8.64 | 12.92 | 5.24 | 4.47 |
| E.% | 8.95 | 13.96 | 13.04 | 12.01 | 4.83 | 0.57 | 21.58 | 0.21 | 5.84 | 14.31 | 17.28 |

Note. Value [1]—reference material: UQAC–FeS, based on certificate; Value [2]—reference material: Mass-1, based on [38]; SD—standard deviation, RSD—relative standard deviation, E—accuracy.

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
