# Peer review of "Rh, Ir, and Ru Partitioning in the Cu-Poor IPGE Massive Ores, Talnakh Intrusion, Skalisty Mine, Russia"

_minerals, doi:10.3390/min12010018_

Round 1
Reviewer 1 Report
The Norilsk-Talnakh deposits are unique geological objects that attract the attention of researchers all over the world. Despite their long-term study, their origin and the processes of ore deposition remain the subject of constant debate.
The manuscript of N.D. Tolstykh and co-authors is devoted to the study of pyrrhotite massive ores from the deepest mine in Eurasia. The authors presented the results of studying the geochemistry and mineralogy of these ores.
There are the following comments to the manuscript:
- The Introduction section is more like a Geological background. There is no analysis of existing ideas about the processes affecting the concentrations and distribution of chalcophilic elements in BMS. There is no clear definition of the aims and objectives of the work.
- When describing the types of the Norilsk-Talnakh ore region, specific terminology is used (see comments in the text, line 47-48), so it makes sense to say this and refer to the article by Barnes et al., 2020 (Barnes, S.J.; Malitch, K.N.; Yudovskaya, M.A. Introduction to a Special Issue on the Norilsk-Talnakh Ni-Cu-Platinum Group Element Deposits. Economic Geology 2020, 115, 1157-1172, doi:10.5382/econgeo.4750.), which provides a detailed explanation of most of the geological terms of Norilsk geologists.
- Figure 1c should show the north-south direction. If it coincides with the vertical frame of the figure, then the cut line on it should have the designation northwest-southeast (NW-SE).
- In Figure 2, the color of intrusive rocks of the Daldykansky complex does not differ from that of disseminated cuprous ores, this introduces confusion.
- In section 2, it would be good to provide a table with a description of the samples studied.
- In the Discussion section, the formation of zonation in pyrites is considered rather one-sidedly. The authors should pay attention to the article by Durand et al., 2015 (Duran, C.J.; Barnes, S.J.; Corkery, J.T. Chalcophile and platinum-group element distribution in pyrites from the sulfide-rich pods of the Lac des Iles Pd deposits, Western Ontario, Canada: Implications for post-cumulus re-equilibration of the ore and the use of pyrite compositions in exploration. Journal of Geochemical Exploration 2015, 158, 223-242, doi:10.1016/j.gexplo.2015.08.002)
Other comments are made in the text of the article.

Reviewer 2 Report
The manuscript "Rhodium, Ru and Ir partitioning in the Cu-poor IPGE massive ores ..." by Tolstykh and others will be interest to most mineralogists and economic geologists particularly those interested in PGE and the Norilsk-Tahakh ore district. In addition, there is considerable useful chemical data and good photomicrograph of ore minerals. There are, however, some changes that should be made to improve the readability. In particular the results and discussion sections are difficult to follow. I had to read the paper three times before I could understand the intended points, but most readers would not be willing to do that. The problem is that the discussion seems like an endless list of overlapping details that are not organized into a clear point by point narrative. I would suggest that each important topic should be separated under secondary headings. Suggested heading tiles include among others: The Talnakh and Kharaelakh intrusions; Immiscible liquids; Geochemistry of ISS and MSS phases; Exsolution of Skalysty ore minerals; and Unusual enrichment of Rh, Ru, and Ir in Skalysty Po. Some line-by-line suggestions follow:
Line 1. Why do you write out the word Rhodium but use the abbreviated Ru and Ir?
Line 2, abstract and conclusions: Although partitioning is the subject of your title, the word "partitioning" is not found anywhere in the abstract or in your conclusion. That needs to change.
Line 3. The title should include the location: Siberia, Russia.
Line 12. Include ", located in Siberia, Russia," after the word "mine"
Line 17. Why "primary" instead of secondary? You need to explain what you mean by primary. You have several stages of fractionation in the text, so it is difficult to keep tract of them. For example, do you mean igneous crystal fractionation or immiscible sulfide melt – silicate melt fractionation.
Line 49. Describe the intrusions, the layering, olivine cumulates, and the location of the massive sulfides. Are they sills, lopoliths, or lacoliths and how thick are they? If they are dikes you can't have a top or base.
Line 66. To avoid any confusion briefly describe each of the fractionation processes that you use, the liquid immiscibility, the development of ISS and MSS, and subsequent solid-state temperature controlled exsolution. Then expand upon these topics one by one.
Lines 85-94. You point out that although most Po ore is poor in PGM, the Po of the eastern flank of the Skalysty mine is enriched in Rh, Ru, and Ir and that this makes the ore unique or atypical. But you never explain how or why.
Lines 95-96. If the geochemistry of the Po ore is a principal objective there should be a clear, concise explanation as to why Rh, Ru, and Ir enrichment has occurred at Skalysty but not elsewhere. This should at least be in the conclusion.
Lines 477-478. Indicate the evidence that IPGE is partitioned into Py.
Line 503. Provide evidence.
Finally, I am a bit surprised that commonly cited and important papers by Ebeil and Naldrett (1994, and 1997), Barnes et al (1985), Ballkaus et al (2001), and Mungall and Brenan (2014) were not cited. But in particular there is an important paper by Patten et al (2013) in Chemical Geology that calculated important partition coefficients (Ds) and reviews partitioning between MSS and ISS. They find that most previous studies have shown that Rh, Ru, and Ir typically partitions into MSS (not rarely) and also undergoes solid state exsolution from the sulfides as PGM during cooling. So, if you agree you need to make clear what you have done that is new. If you disagree with any part of that you need to explain why.
Reviewer 3 Report
Dear authors and editor,
The paper by Tolstykh et al shows a significant finding - that magmatic pyrite concentrates PGE. This is a surprising result and worth of publication.
However, the papr requires some work.
See attached file has many small comments to address.
I also think the paper could use a bit of context with the scientific literature on PGE from outside Norilsk or Russia, which are currently strongly over-represented in the citations.
I think that an important paper to consider regarding the magmatic crystallisation textures is probably this one:
https://doi.org/10.1007/s00410-021-01868-4
How do your textures compare to theirs? I see many similarities. Are their explanations consistent with yours? What about the magmatic pyrite?
You have lots of arsenides, which you do not consider in your discussion. Some works to check:
https://doi.org/10.1007/s00410-013-0951-9
https://doi.org/10.2138/am-2020-7477
https://doi.org/10.1007/s00410-020-01705-0
https://doi.org/10.1038/ncomms3405
You say that the PGE are in solid solution in pyrite. Is this the case? Could it be an artifact of very small nuggets, on the nanoscale? This option has not been considered at all. I suggest to refer to these two papers and show this as a possibility in your work, as the PGE may have simply been captured as discrete nanonuggets while the system was still liquid, and physically "stuck" to crystallising pyrite instead of chemically partitioned. See these three papers:
https://doi.org/10.1130/G47086.1
https://doi.org/10.2138/am-2018-6424
https://doi.org/10.1130/G47579.1
And in general I would think that a quick comparison between your study and similar studies of other places in the world could be useful:
https://doi.org/10.1016/j.oregeorev.2020.103339
https://doi.org/10.1007/s00126-020-01033-0
https://doi.org/10.3749/canmin.2000100

Round 2
Reviewer 1 Report
As we have already noted earlier, the work is of great interest for understanding the processes of ore formation. It presents new original information about the unique ores of the Norilsk region.
The authors made significant corrections to the text, took into account all comments. The manuscript can be published in a special issue of the journal Minerals.
Author Response
Thanks to the distinguished reviewer for revisiting our manuscript and agreeing to submit it to Minerals
Reviewer 2 Report
The manuscript "Rh, Ru and Ir partitioning ..." by Nadezhda Tolstykh and others has been sufficiently revised to the point where it is now publishable in my opinion. Almost all of the confusion that I found in the original draft has been eliminated so that it is now clear, interesting, and useful.
Author Response

(The authors gave the same response as above.)

Reviewer 3 Report
Only some minor corrections for clarity this time:
line 50: change "through" to "trough"
line 123: You use the term BMS here, but only define it in the next paragraph. Move the definition to here please.
line 124: I don't think that "Alternatively" is required here?
line 156: But the JXA-8100 is also an EPMA, and you're writing this paragraph as if it's not? These elements were also measured on WDS, right?
line 163: 0.2 seconds for what? Each pixel in the map? Please clarify.
line 184: "analyses"
line 186: That's not the ablation speed, it's the speed of moving the laser beam on top of the sample surface, right?
line 190: "EPMA" - use the term you defined earlier.
line 201: "analyses"
line 237: This is a bit unclear. Are the magnetites themselves zoned, or are they surrounded by a zone of ilmenite? The figures make it seem like it is the latter.
line 255: Which "sulfide occurrences"?
line 273: Is Fe3NiS4 a new undescribed mineral, or a different stoichiometry of a known mineral?
line 320: Is there a reference showing this to happen experimentally, or otherwise?
line 344: What's 0.38%? Up to that amount? Average amount?
line 354: You're already using atomic percent, no need to change to permille notation here. Keep to percent.
table 1,3,4: Instead of having a table footnote for the mineral names, why not include the mineral names as another column in the table itself? It would be much more readble.
fig 10: Change the colours here - red-green is the most common colour blindness. I recommend changing one of Pt or Pd to blue. Or use the same style you used in fig 12 later.
line 426: "...the sum of which may average up to 28 ppm"
table 5: "...in rock and tenor in 100% sulfide,..." - is this what you mean?
table 6: Use the multiplication symbol × instead of asterisk (*) symbol or x letter.
table 7: What's "D"? What does "macrocomponent" mean?
line 468: Use the multiplication symbol × instead of x letter. Here and elsewhere.
Author Response
We are very grateful to the reviewer for re-proofreading our manuscript.
